# Evaluating enhanced sampling rate for turbulence measurement with wind lidar profiler

Maxime Thiébaut [1], Louis Marié [2], Frédéric Delbos [3], and Florent Guinot [1]

[1]France Énergies Marines, Technopôle Brest-Iroise, 525 Avenue Alexis de Rochon, 29280 Plouzané, France
[2]Laboratoire d'Océanographie Physique et Spatiale, Université de Brest, CNRS, IFREMER, IRD, Plouzané, France
[3]Vaisala France SAS, 6A, rue René Razel, Tech Park, CS 70001, 91400 Saclay Cedex, France

**Correspondence:** Maxime Thiébaut (maxime.thiebaut@france-energies-marines.org)

**Abstract.** This study evaluates the impact of an enhanced sampling rate on turbulence measurements using the Vaisala WindCube v2.1 lidar profiler. A prototype configuration, sampling four times faster than the commercial setup, was compared to the commercial WindCube v2.1 with reference measurements provided by a 2D sonic anemometer mounted on a measurement mast. Over the 47-day experiment, the prototype configuration showed performance similar to the commercial setup for key performance indicators (KPIs) like slope and coefficient of determination of mean wind speed compared to reference measurements, with both configurations meeting "best practice" threshold. However, for mean wind speed differences, the commercial configuration met the "best practice" level, while the prototype met the "minimum acceptance" criterion. Additionally, the data availability of the prototype configuration was 0.5% lower than that of the commercial configuration. Moreover, the increased sampling rate in the prototype lidar resulted in higher mean variance in instrumental noise compared to the commercial configuration. Despite this limitation, the mean noise-corrected along-wind variance measured by the prototype lidar was approximately 7% higher than that of the commercial lidar. This effect was especially evident at higher wind speeds. Error metrics for the noise-corrected along-wind standard deviation in the prototype lidar were approximately 25% lower than those of the commercial configuration. However, the observed improvements of the prototype configuration in measuring turbulence fell short of expectations due to inherent limitations in the measurement process within the probe, where spatial and temporal filtering effects constrain the detection of turbulence at certain scales.

## 1 Introduction

Accurate turbulence data enable better understanding and control of flow patterns, optimizing the design, operation, and maintenance of wind energy systems. Turbulence plays a critical role in the wind energy sector because it directly influences the unsteady aerodynamic loads experienced by turbines (e.g., Frandsen, 2007; Mücke et al., 2011; Dimitrov et al., 2017). These fluctuating loads affect the structural integrity and operational stability of wind turbine components. Therefore, precise turbulence measurement is essential for enhancing the efficiency and safety of turbine operations, minimizing wear and tear on vital components, and extending the lifespan of these costly assets. Additionally, improved turbulence characterization can facilitate more precise wind resource assessments, aiding in site selection and the overall planning of wind energy projects (e.g., Yang et al., 2021).

In the wind energy sector, the utilization of wind lidar profiler technology has gained significant traction in recent years, complementing the traditional meteorological mast equipped with in-situ sensors like cup or sonic anemometers as the standard means of measuring key mean wind properties, such as speed and direction. Wind lidar profilers present compelling advantages, including the potential for cost reduction compared to meteorological masts and the capacity to measure at similar or even greater heights above the ground (e.g., Gottschall et al., 2012).

Wind lidars profilers can be categorized according to their emission waveform, i.e., pulsed or continuous, and measuring technique, i.e., Doppler beam swinging (DBS) (Strauch et al., 1984) or velocity-azimuth display (VAD) (Browning and Wexler, 1968). Measurement methods used by wind lidar profilers are fundamentally different from those used by cup or sonic anemometers. Anemometers estimate wind speed over a small volume of just a few cubic centimeters, whereas pulsed lidar profilers provide an average over a cylindrical probe several dozen meters long with a cross-sectional diameter of less

than 1 cm (Fig. 1).

    However, wind lidar profilers have yet to garner widespread acceptance for turbulence measurement, which remains a focal point of ongoing research. In contrast to turbulence data derived from reference instruments such as sonic anemometers, turbulence data from lidar profiler measurements suffer from systematic errors induced by (i) the inter-beam effect, also known

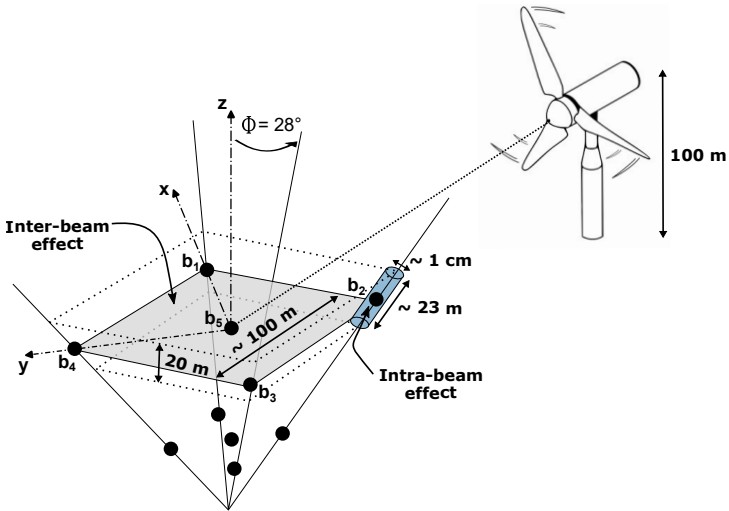

**Figure 1.** A schematic illustration of inter- and intra-beam effects in the WindCube v2.1 lidar profiler measurement process. The blue cylinder represents the probe volume, corresponding to the dimensions of the commercial lidar configuration. The positions of the five beams are labeled as $b_i$, where $i$ ranges from 1 to 5. The inclination of the diverging beams (from beam 1 to beam 4) with respect to the vertical $z$-axis is $\phi = 28°$. Beam 5 is aligned with the $z$-axis, while beams 1 and 3 are aligned with the $x$-axis, and beams 2 and 4 are aligned with the $y$-axis in the coordinate system of the instrument, as stipulated by the manufacturer. The black dots indicate the centers of the probe measurement volumes.

as the cross-contamination effect, (ii) the intra-beam effect, i.e., the space-time averaging effect within the probe volume (Fig. 1) and, (iii), instrumental noise.

The inter-beam effect can result in either underestimation or overestimation of turbulence metrics, arising from the modulation of energy associated with eddies of specific wavenumbers (Kelberlau and Mann, 2020). This effect is particularly relevant in the context of the assumption of instantaneous homogeneity, which underlies multi-beam lidar measurement techniques. Under this assumption, the turbulent field is considered spatially homogeneous across the beams at each instant in time, a condition that, if violated, can lead to inter-beam contamination. Any phase difference between the horizontal and vertical components of an eddy significantly impacts the filtering of flow structures, potentially leading to both amplification or attenuation of their measured turbulent energy (Theriault, 1986; Gargett et al., 2009).

The intra-beam effect refers to a probe-time averaging phenomenon occurring within the lidar probe, leading to an underestimation of turbulence metrics. It arises from two anisotropic filtering processes: (1) spatial filtering due to averaging over the probe volume and (2) temporal filtering caused by averaging over the beam's pulse accumulation time, $\Delta t$, at a given measurement position. These two effects give rise to a transfer function, $H$, applied by the instrument on the signal measured within the probe. The transfer function includes a part due to time-averaging (the sinc term) and a part due to space-averaging (the Gaussian term), such that:

$$|H|^2(\mathbf{k}) = \text{sinc}^2\left(\frac{\Delta t}{2}\mathbf{k}\cdot\mathbf{U}\right)\exp\left(-\left[\sigma_l^2(\mathbf{k}\cdot\mathbf{b})^2 + \sigma_r^2(\|\mathbf{k}\|^2 - (\mathbf{k}\cdot\mathbf{b})^2)\right]\right) \tag{1}$$

Here, $\mathbf{k}$ is the turbulent structure wavevector, $\mathbf{b}$ is the beam pointing vector, $\mathbf{U}$ is the vector associated with the wind direction of magnitude $U$, and $\sigma_l$ and $\sigma_r$ represent the Gaussian weighting factors in the along-beam and cross-beam directions, respectively. A detailed mathematical derivation of Eq. 1 is provided in the supplementary material.

From Eq. 1, it follows that wind field structures with wavelengths smaller than $\sigma_l$ in the along-beam direction are attenuated, as are those with wavelengths smaller than $\sigma_r$ in the cross-beam direction. However, in the latter case, these structures are so small that the filtering effect becomes negligible, as the cross-section of the probe is approximately 1 cm (Fig. 1). Ultimately, assuming the Taylor frozen turbulence hypothesis, the wavevector domain that passes through the filter is defined by the intersection of two slices: one perpendicular to the direction of $\mathbf{U}$, which preserves structures longer than $\pi\Delta tU$, and another perpendicular to the direction of $\mathbf{b}$, which retains structures longer than $\sigma_l$. All other structures are filtered out.

Pulsed lidar profilers require several seconds to complete a full scanning cycle resulting in a low sampling rate that causes discrepancies between turbulence measurements taken by anemometers and those by lidar profilers (e.g., Peña et al., 2009). While the sampling rate governs how quickly the lidar progresses through a scan cycle, it is directly influenced by pulse accumulation time. Consequently, even if the sampling rate is increased, pulse accumulation can still limit the ability of the lidar to resolve small-scale turbulent structures. Since turbulent motion scales vary from milliseconds to hours and from centimeters to kilometers (e.g., Stull, 2000), it is crucial to account for both temporal and spatial filtering effects when assessing lidar-based turbulence measurements.

The concept of measuring turbulence using remote sensing instruments has gradually evolved since the early works in radar meteorology by Lhermitte (1962) and Browning and Wexler (1968). Lhermitte (1969) was the first to propose a method for inferring turbulence by analyzing the variance of radial velocity measurements through VAD scanning. Following this, Wilson (1970) conducted pioneering experiments using a pulsed Doppler radar to detect turbulence within the convective boundary layer (0.1-1.3 km). However, these early measurements were limited to turbulence scales larger than the pulse volume and smaller than the scanning circle, and no validation against reference instruments was performed, questioning their reliability.

Kropfli (1986) expanded Wilson's approach to capture turbulence scales larger than the scanning circle by integrating data from multiple scans. Although initially developed for Doppler radar, these methods were later adapted for Doppler lidar. Eberhard et al. (1989) were the first to apply Wilson's and Kropfli's methods using lidar, and Gal-Chen et al. (1992) further refined the technique with a different scanning configuration. Despite these advancements, the significant probe length (around 100 m) limited studies to the convective boundary layer due to considerable probe volume averaging, especially near the ground. Frehlich (1994) and Frehlich et al. (1994) demonstrated the averaging effect in the measurement of the structure function, showing that this effect becomes more pronounced at smaller separation distances. To address this limitation, research shifted toward understanding and mitigating probe volume averaging effects. Smalikho et al. (2005) provided explicit formulae to account for the small-scale filtering effect of the finite probe volume in continuous-wave lidar, proposing three different methods for a staring lidar: (1) using the width of the Doppler spectrum, (2) the velocity structure function, and (3) the one-dimensional velocity spectrum. The expression for the structure function was derived under the assumption of local isotropy in the inertial subrange. Kristensen et al. (2011) later re-derived this expression, assuming a Lorentzian probe volume weighting function. However, averaging effects continue to pose challenges for turbulence measurements in the surface layer, where wind turbines operate. These effects lead to an underestimation of variance derived from wind lidar compared to reference turbulence measurements, as demonstrated in (e.g., Mann et al., 2009; Sjöholm et al., 2009).

This paper explores advancements in the Vaisala WindCube v2.1 lidar profiler, focusing on a key modification: increasing the sampling rate by reducing the pulse accumulation time. This enhancement is assessed for its impact on measuring mean wind speed, data availability, and along-wind variance and its square root, i.e., the standard deviation. The latter is particularly important, as it is used in the wind power industry to compute turbulence intensity (TI), a critical metric for turbine load assessment, site suitability, and energy yield predictions. Additionally, the influence of instrumental noise is analyzed to confirm that the potential improvements in turbulence estimates with the higher sampling rate are not simply a result of increased noise.

The paper begins with a detailed overview of the data and methods, including the prototype configuration of the WindCube v2.1 and the field measurement setup (Section 2). The study compares the prototype configuration against the commercial WindCube v2.1 and a sonic anemometer installed on a meteorological mast, which serves as the reference measurement. The methodology section then focuses on velocity spectra analysis, instrumental noise evaluation, and variance computation in instrument coordinates. Additionally, key performance indicators and error statistics used for validation are outlined. The results section presents findings on mean wind speed, data availability, standard deviation, variance and instrumental noise contributions (Section 3). This is followed by a discussion of the implications, addressing both the advantages and challenges

of a higher sampling rate (Section 4). The paper concludes with key takeaways on how the increased sampling rate enhances turbulence detection while considering measurement limitations and filtering effects (Section 5).

## 2 Data and methods

### 2.1 Prototype configuration with increased sampling rate

The WindCube v2.1 lidar is designed for general atmospheric measurements, such as mean wind speed and direction, requiring
a careful balance between temporal resolution, spatial resolution, and carrier-to-noise ratio (CNR). Its default sampling rate is optimized to ensure high data quality and availability across varying altitudes and atmospheric conditions while maintaining system efficiency and manageable data processing.

The WindCube v2.1 employs the Doppler Beam Swinging (DBS) technique to measure wind speed. This method utilizes an optical switch that sequentially directs the lidar beam toward four directions (0°, 90°, 180°, and 270° relative to the reference
$x$-axis), each inclined at $\phi = 28°$ from the vertical. A fifth beam is directed vertically upwards, resulting in wind measurements at five distinct positions (Fig. 1-2).

In its standard commercial configuration, the WindCube lidar collects data at each position for approximately $\Delta t = 0.8$ s before switching to the next. Including transition times, a complete DBS scan is performed in 4 s, yielding a line-of-sight (LOS) velocity sampling rate of 0.25 Hz (Table 1). This sampling rate is well-suited for capturing turbulent structures larger
than 100 m. However, wind turbine components experience loads from turbulence across a wide range of scales. Increasing the

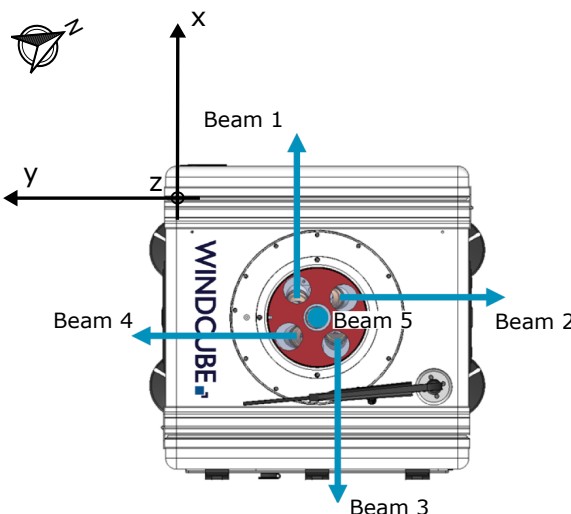

**Figure 2.** Top view of a WindCube v2.1 lidar showing the positions of its five beams. The $x$-axis is oriented from beam 3 towards beam 1, the $y$-axis extends from beam 4 towards beam 2, and the vertical $z$-axis points upward along beam 5. The arrow indicates North. For the present study, the primary $x$-axis of the lidars was oriented at -62° relative to North.

**Table 1.** LOS velocity measurement parameters for the commercial and prototype WindCube v2.1 configurations.

| Configuration | LOS Sampling Rate (Hz) | Accumulation Time (s) | LOS Samples per 30 min | Probe Length (m) |
|---|---|---|---|---|
| Commercial | 0.25 | 0.8 | 450 | 23 |
| Prototype | 1.00 | 0.2 | 1800 | 23 |

sampling rate is crucial for broadening the velocity spectrum captured by the lidar, potentially enabling the detection of extra turbulent energy that influences turbine performance.

Theoretically, a higher sampling rate improves temporal resolution and extends the resolved turbulence frequency range. However, for wind lidar profiler technology, this enhancement comes with trade-offs. The duty cycle, which represents the proportion of time the lidar transmits pulses, decreases as sampling rate increases, potentially reducing signal strength. Moreover, increasing the sampling rate requires a reduction in accumulation time, resulting in fewer pulses per sample and increasing noise. The commercial WindCube v2.1 configuration balances these factors to maximize data reliability. It integrates a high number of pulses per measurement to enhance signal quality, making it well-suited for general wind resource assessment. However, its probe length of approximately $L_{probe} = 23$ m (Fig. 1, Table 1) limits its ability to resolve small eddies compared to point sensors like sonic anemometers.

In response to the demand for capturing additional turbulent energy, we developed a modified version of the WindCube v2.1 that operates four times faster, achieving a LOS velocity sampling rate of 1 Hz. This modification was achieved by reducing the accumulation time for data collection from each beam in conjunction with a reduction in the number of transmitted pulses. The factor of 4 was chosen as a compromise between increasing temporal resolution and maintaining an acceptable CNR and data availability. This choice is intended to keep wind measurements comparable to those from the commercial configuration while enabling the capture of additional turbulent energy. The actual impact on measurement performance will be assessed in the study.

## 2.2 Field measurement

The field measurement campaign was conducted by DNV at the lidar validation test site in Janneby, Germany (Fig. 3). The site's flat terrain ensures orography-free flow, making it ideal for lidar verification trials. It offers good exposure to largely undisturbed wind conditions from most directions. Situated just a few meters above mean sea level, the site features low surface roughness due to its predominantly agricultural land use (Fig. 3a). Two wind turbines (WT N100 and WT N117; Fig. 3a) are located near the meteorological mast. Their wake-affected wind sectors are shown in blue in Fig. 4 and lie outside the sectors selected for turbulence analysis (gray areas in Fig. 4, Section 2.5). The closest turbine is 210 m from the mast. A few small human-made structures (e.g., houses, sheds), all under 15 m in height, are situated about 500 m southwest of the mast.

The meteorological mast is a 100 m, 3-fold guyed lattice tower with a constant face width of 0.4 m. It is equipped with six MEASNET-calibrated Thies First Class Advanced cup anemometers (No. 4.3352) and a Thies 2D sonic anemometer (No.

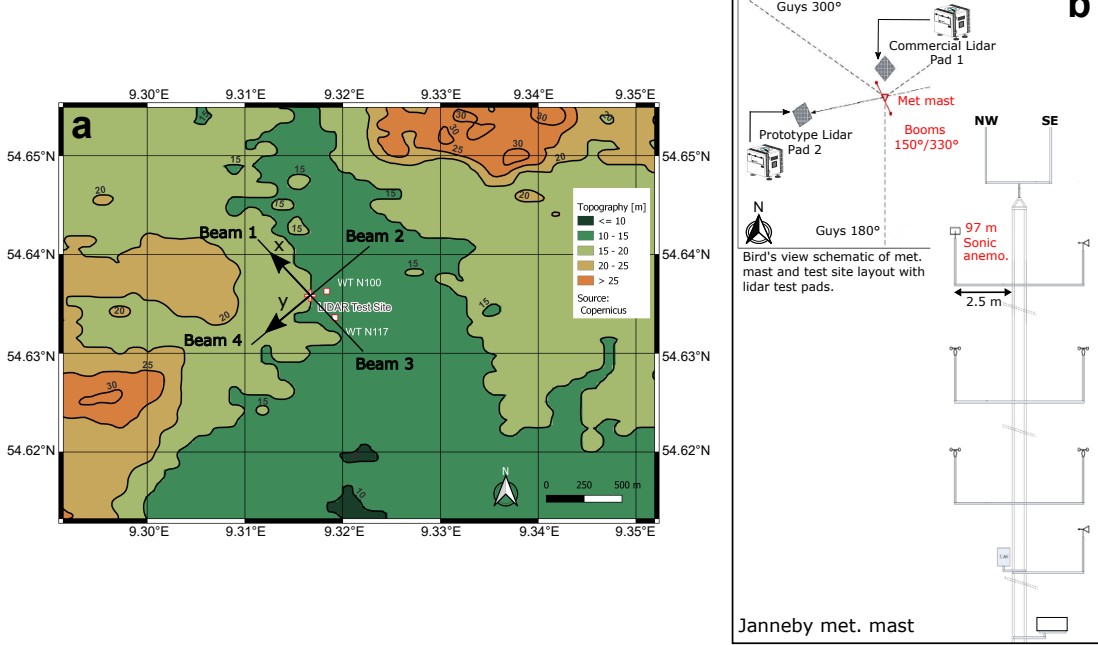

**Figure 3.** a: Test site location at Janneby, Germany. Black arrows indicate the beam orientations for the commercial and prototype configurations. The $x$ and $y$ axes of the instrument coordinate system (see Fig. 2) are marked with black arrows. b: Configuration of the meteorological mast, showing the position of the sonic anemometer. NW and SE denote the north-west and south-east directions. The schematic in panel b also provides a bird's-eye view of the meteorological mast and test site layout, including the lidar test pads.

4.3830). However, only the Thies 2D sonic anemometer is used in this study to provide reference measurements of mean
wind speed and turbulence, as the cup anemometers data are not available. The mounting arrangements are consistent with the currently valid IEC and IEA recommendations for the use of anemometry at meteorological masts. As shown in Fig. 3b, the sonic anemometer is pointing towards 150° from True North and is mounted at 97 m above ground, which corresponds to the average hub height of modern land-based wind turbines. The sonic anemometer was set to record continuous horizontal wind speed and direction at sampling rate of 4 Hz. Potential wake effects from the meteorological mast structure were considered
in the analysis of the sonic anemometer data. The wind directions associated with flow disturbances caused by the mast itself overlap with the wake sector of wind turbine WT N117 which was excluded from the analysis.

    Adjacent to the measurement mast, both the commercial lidar configuration, and a prototype version with an enhanced sampling rate were installed 3 m and 13 m apart the mast respectively. Both lidars were aligned such that beams 1 and 3, which correspond to the $x$-axis (Fig. 2), were oriented at -62° from True North (Fig. 3). According to the manufacturer's
recommendation, the $x$-axis is the primary axis and should be oriented relative to North. Beams 2 and 4 are fixed along the $y$-axis. One of the measurement heights of both lidars was set to 97 m above ground to coincide with the height of the sonic anemometer deployment on the mast.

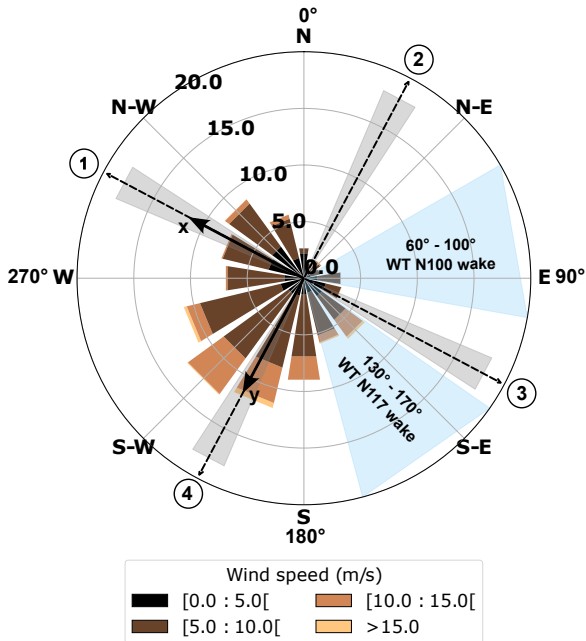

**Figure 4.** Wind rose showing wind data recorded over 47 days by the sonic anemometer at 97 m above ground level. Gray shaded areas indicate the wind sectors selected for the turbulence analysis in this study, corresponding to events when the wind was aligned ($\pm$ 5°) with either the beam pair 1-3 (aligned with the $x$-axis) or beam pair 2-4 (aligned with the $y$-axis), as numbered in circles. Blue shaded areas indicate wind sectors contaminated by the nearby wind turbines WT N100 and WT N117 (Fig. 3a).

The field measurement campaign was conducted over two periods: from 12 to 25 November 2021, and from 07 December 2021 to 10 January 2022. These two measurement periods were combined to form a 47-day dataset. To facilitate a comparison of turbulence measurements, the sonic-derived wind dataset was resampled to match the sampling rate of the LOS velocities measured by the prototype configuration. This ensures that similar turbulence time scales are captured when calculating and comparing turbulence estimates. Therefore, the sonic anemometer measurements were resampled at 1 Hz.

The 47-day dataset was segmented into 2,256 subsets, each comprising 30-min of data. For each subset, the commercial lidar provided 450 measurement points, while the prototype lidar provided 1,800 points, corresponding to their respective sampling rates of 0.25 Hz and 1 Hz (see Table 1). The selection of a 30-min window, rather than the standard 10-min interval commonly used in the wind energy industry, was guided by the aim of reducing random errors in turbulence measurements, following the recommendations of Lenschow et al. (1994).

### 2.3 Velocity spectra

Power spectral density of the velocity, i.e., the velocity spectra, provide valuable information about the distribution of turbulent kinetic energy across different scales of motion within the wind flow. This understanding helps in characterizing turbulence and its effects on wind turbine performance and structural loads.

Velocity spectra were computed using Welch's method (Welch, 1967). This method computes an estimate of the spectrum by dividing the data into overlapping segments, computing a modified periodogram for each segment and averaging the periodograms. The Hann window with 50% overlap was applied to each segment to reduce spectral leakage and improve frequency resolution. The 50% overlap is a reasonable trade off between accurately estimating the signal power, while not over counting any of the data.

Fitting turbulence velocity spectra derived from lidar-reconstructed velocity components to turbulence models should be avoided due to the inter-beam effect, which distorts the spectra and complicates their physical interpretation. Therefore, such spectra were not considered in this study. The focus was on velocity spectra $S_i(f)$ derived from the LOS velocities measured by beam $i$. The primary limitation in this approach is the intra-beam effect. Spectra were computed for each 30-min subset of data.

The spectra, $S_i(f)$, were fitted by a parametric expression (Teunissen, 1980; Olesen et al., 1984; Tieleman, 1995) in the frequency domain $f$, to which we add a component $N_i$ associated with the power spectral density of instrumental noise of the LOS velocity measured by beam $i$ (see section 2.4):

$$S_i(f) = \frac{m}{(1 + nf)^\beta} + N_i \tag{2}$$

The coefficient $m$ primarily controls the vertical scaling or amplitude of the spectum whereas $n$ influences the rate at which the function decays as $f$ increases. The exponent $\beta$ determined the shape of the spectrum.

Three different weighting schemes were considered: an unweighted scheme, a low-frequency weighted scheme with weights proportional to the logarithm of the frequency, and a high-frequency weighted scheme with weights inversely proportional to the logarithm of the frequency. Assessing the fitting accuracy included comparing the variance obtained from the integrated fitted spectra with the measured spectra, and calculating their absolute relative differences.

## 2.4 Instrumental noise

Lidar measurements are inherently influenced by signal noise and potential variations in aerosol fall speeds, both of which contribute additional terms to the observed variance. Assuming that all atmospheric flow contributions to the observed LOS velocity variance within the considered short timescales are of a turbulent nature, the variance $\sigma_{b_i}^2$ of the LOS velocity measured by beam $i$, can be expressed as the sum of three independent terms (Doviak and Zrnic, 1993):

$$\sigma_{b_i}^2 = \sigma_{p_i}^2 + \sigma_{n_i}^2 + \sigma_{d_i}^2 \tag{3}$$

Here, $\sigma_{p_i}^2$ represents the net contribution from atmospheric turbulence at scales measurable by the lidar (Brugger et al., 2016), $\sigma_{n_i}^2$ denotes the variance associated with instrumental noise, and $\sigma_{d_i}^2$ accounts for the variance caused by variations in aerosol terminal fall speeds within the probe volume. However, $\sigma_{d_i}^2$ can typically be neglected, as particle fall speeds are generally less than 1 cm/s (e.g., Bodini et al., 2018). Noise has been identified through two different methods: a spectral approach and an autocorrelation approach, as accurately identifying the variance of noise is critical to our study.

### 2.4.1 Spectral method

Instrumental noise is a critical factor in the spectral analysis of velocity time series. In the spectrum of a velocity time series, this noise typically manifests as a flattening of the spectrum at higher frequencies, indicating a white noise characteristic that contributes equally across these frequencies (e.g., Thomson et al., 2012; Durgesh et al., 2014; Guerra and Thomson, 2017; McMillan and Hay, 2017; Thiébaut et al., 2020). At lower frequencies, the spectrum is usually dominated by the actual signal, which may show a characteristic decay or specific features related to the physical process being measured, such as turbulence. As frequency increases, the influence of the instrumental noise becomes more prominent, leading to a flattened spectral region where the noise dominates.

In Eq. 2, $N_i$ represents the constant power spectral density of noise, which contributes to the spectral flattening observed at higher frequencies. The variance of the noise depends on the technical characteristics of the device measuring the velocity, such as Nyquist velocity, the signal spectral width, the number of pulses and points per range gate, and the signal-to-noise ratio. Theoretical expressions for the variance of this noise can be derived and subsequently removed from the computed turbulence metrics to improve accuracy (Pearson et al., 2009; O'Connor et al., 2010; Bodini et al., 2018, 2019; Wildmann et al., 2019). However, the technical specifications of lidar profilers are no longer openly shared with users, making it impossible to evaluate this noise theoretically. To address this, it is essential to evaluate the noise using an alternative method, such as the spectral approach employed in this study. This approach is comparable to the method proposed by (e.g. Richard et al., 2013; Durgesh et al., 2014). It enables the determination of the power spectral density of noise, $N_i$, associated with the LOS velocity measured by beam $i$. Subsequently, the variance of the instrumental noise, $\sigma_{n_i}^2$, can be derived by multiplying $N_i$ by the Nyquist frequency, $f_N$, such as (e.g., McMillan and Hay, 2017):

$$\sigma_{n_i}^2 = N_i f_N \tag{4}$$

### 2.4.2 Autocorrelation function method

An alternative method for computing the variance of the instrumental noise involves the calculation of the auto-correlation function (ACF) of the squared LOS velocity time series, as proposed by Lenschow et al. (2000). The ACF quantifies the similarity between a signal and its time-shifted versions across various time lags. This measure provides insight into how much of the signal correlates with its past values, which is essential for distinguishing between the noise and signal components.

According to Lenschow et al. (2000), after calculating the ACF, the ACF values (excluding the first lag) are fitted to a 2/3 power-law function. This power-law model describes the decay of correlation over time, allowing for the extraction of a coefficient that characterizes how the correlation diminishes as the time lag increases. From this power-law fit, the value of the ACF as the lag tends to zero is estimated by extrapolation of the fitted model. This value is associated with the signal variance.

Subsequently, the total variance of the signal is calculated. The instrumental noise variance, $\sigma_{n_i}^2$, is then determined by subtracting the signal variance, as derived from the fitted power-law model, from the total variance. This process enables the separation of the signal and noise contributions to the overall variance. The application of the ACF method requires that the data be stationary. To verify this assumption, a stationarity test was performed for each 30-min subset using the Augmented

Dickey-Fuller (ADF) test. The null hypothesis of the ADF test states that the data are non-stationary which was assessed through the determination of a significance level, commonly set at 0.05 or 5% (e.g. Hayat, 2010).

## 2.5 Computation of the variance in instrument coordinates

The conventional method for computing variance and standard deviation (the square root of variance) from wind lidar profiler measurements relies on deriving second-order statistics from the reconstructed instantaneous velocity components based on LOS velocities. This approach inherently combines, at each time step, measurements taken at sampling points separated by several tens of meters, depending on the height level of interest. The assumption of instantaneous flow homogeneity (inter-beam effect) introduces an uncertainty in the derived statistics, which is difficult to quantify and can lead to either an overestimation or underestimation of the standard deviation, depending on the frequency and flow configuration. Additionally, this traditional method is affected by both intra-beam filtering and instrumental noise. Crucially, because variance is computed from the reconstructed instantaneous velocity components, it does not account for the noise-induced variance present in the LOS velocity time series which will result in overestimation of variance.

The combined influence of the inter-beam effect, intra-beam effect, and instrumental noise can result in variance estimates derived from the traditional approach that may appear to align more closely with those derived from a sonic anemometer, but for reasons unrelated to the actual turbulence characteristics. Consequently, the benefits of an increased sampling rate for turbulence measurement using a lidar profiler cannot be accurately assessed with this approach.

The variance method, as referred to in the studies (e.g., Stacey et al., 1999a, b; Lu and Lueck, 1999; Rippeth et al., 2002; Guerra and Thomson, 2017; Thiébaut et al., 2022), offers an alternative to the traditional approach for computing variance. This method calculates the second-order statistics of the three velocity components by deriving them directly from the second-order statistics of the LOS velocities. Unlike the traditional approach, the variance method is unaffected by the inter-beam effect. However, it is still influenced by the intra-beam effect and instrumental noise. Notably, the impact of instrumental noise can be identified and removed. Hereafter, a hat notation is used to denote standard deviation or variance derived from this method.

The variance method enables the calculation of the variances, $\hat{\sigma}_x^2$ and $\hat{\sigma}_y^2$ of the velocity components $u_x$ and $u_y$ (in instrument coordinates) as:

$$\hat{\sigma}_x^2 = \frac{1}{2\sin^2\phi}\left(\sigma_{p_3}^2 + \sigma_{p_1}^2 - 2\cos^2\phi\,\sigma_{p_5}^2\right) \tag{5}$$

$$\hat{\sigma}_y^2 = \frac{1}{2\sin^2\phi}\left(\sigma_{p_2}^2 + \sigma_{p_4}^2 - 2\cos^2\phi\,\sigma_{p_5}^2\right) \tag{6}$$

where $\sigma_{p_i}^2 = \sigma_{b_i}^2 - \sigma_{n_i}^2$ (Eq. 3), is the variance of the LOS velocity recorded by beam $i$, corrected for the variance of instrumental noise.

In this paper, we restrict the application of the variance method to situations where the wind aligns ($\pm$ 5°) with a single pair of opposite beams (either pair 1-3 or pair 2-4) of the lidar profilers. This alignment condition was met in 17.1% of the cases (gray areas in Fig. 4). Under these conditions, it can be reasonably assumed that the covariance term, $\hat{\sigma}_{uv}$ (where $v$

represents the cross-wind velocity), which corresponds to $\hat{\sigma}_{xy}$ in this specific condition, is negligible (e.g., Newman et al., 2016). Specifically, when the wind aligns with beams 1 and 3, we have $\hat{\sigma}_u^2 = \hat{\sigma}_x^2$. Conversely, when the wind aligns with beams 2 and 4, it follows that $\hat{\sigma}_u^2 = \hat{\sigma}_y^2$. For brevity, we use $\hat{\sigma}^2$ in place of $\hat{\sigma}_u^2$ hereafter. The standard deviation, $\hat{\sigma}$, is then compared to the along-wind standard deviation, $\sigma$, which is derived from sonic anemometer measurements.

## 2.6 Key performance indicators and acceptance criteria

The first step of our analysis focuses on key performance indicators (KPIs) related to mean wind statistics, such as wind speed. These include mean differences, slope, and the coefficient of determination ($R^2$) at reference heights corresponding to sonic anemometer measurements. The verification process follows standard lidar performance requirements set by DNV (2009), which define acceptance criteria (ACs) as either "best practice" or "minimum allowable tolerances." Applied to wind speed, these criteria flag any KPIs exceeding the defined thresholds as "deviations." Table 2 summarizes the ACs established by DNV, which are tested in this paper for wind speed KPIs.

Additionally, the paper addresses data availability. Data availability is defined as the ratio of valid data points returned by the lidar to the maximum number of possible points that could be acquired during the test. To pass the test, the standard lidar performances set the data availability threshold at 90% (Table 2).

## 2.7 Error statistics metrics

This paper focuses on turbulence measurements, specifically the standard deviation and variance of the along-wind velocity, obtained from both the commercial and prototype lidars. These measurements are compared to the standard deviation provided by the reference instrument: the sonic anemometer. To assess the accuracy and reliability of the lidar turbulence measurements, various error statistics are used. These include:

**Table 2.** Acceptance criteria for KPI of mean wind speed in wind lidar profiler certification.

| KPI - Wind speed | Definition | Best practice | Minimum | Deviation |
|---|---|---|---|---|
| Difference | Percentage difference in mean wind speeds between lidar and reference over the verification campaign, relative to the campaign mean wind speed. | < 1% | [1–1.5]% | > 1.5% |
| Slope | Slope from single-variable regression, constrained to pass through the origin. | [0.98–1.02] | [0.97–1.03] | < 0.97 or > 1.03 |
| $R^2$ | Correlation coefficient from single-variable regression. | > 0.98 | > 0.97 | ≤ 0.97 |
| Data availability | Mean percentage of available data points in each 30-min subset, relative to the total number of possible records. | ≥ 90% | - | < 90% |

– **Root Mean Square Error (RMSE)**: quantifies the average magnitude of the errors.

$$\text{RMSE} = \sqrt{\frac{1}{M}\sum_{i=1}^{M}(X_i - Y_i)^2} \tag{7}$$

– **Mean Absolute Error (MAE)**: calculates the average absolute difference between predicted and observed values.

$$\text{MAE} = \frac{1}{M}\sum_{i=1}^{M}|X_i - Y_i| \tag{8}$$

– **Bias**: represents the systematic error between the lidar and reference measurements.

$$\text{Bias} = \frac{1}{M}\sum_{i=1}^{M}(X_i - Y_i) \tag{9}$$

– **Coefficient of determination ($R^2$)**: indicates the proportion of variance in the lidar measurements explained by the reference data.

$$R^2 = 1 - \frac{\sum_{i=1}^{M}(X_i - Y_i)^2}{\sum_{i=1}^{n}(Y_i - \bar{Y})^2} \tag{10}$$

Here, $X_i$ represents the lidar measurement, $Y_i$ the corresponding reference measurement from the sonic anemometer, $\bar{Y}$ is the mean of the reference measurements, and $M$ is the number of turbulence estimates. Together, these statistical metrics provide a comprehensive evaluation of the lidar's performance in capturing turbulence characteristics relative to the reference instrument.

## 3 Results

### 3.1 Mean wind speed and data availability

The first step in proposing enhancements to lidar technology is to evaluate their impact on mean wind speed measurements. Fig. 5a illustrates that the mean vertical wind speed profiles measured by both configurations are closely aligned. However, the difference between the mean wind speed measurements provided by the commercial configuration and the reference measurement (black cross in Fig. 5) at the reference altitude is smaller, amounting to 0.98%, compared to a 1.41% difference for the prototype configuration. These results demonstrate that the commercial configuration closely matches the "best practice" AC criterion for the difference in mean wind speed, while the prototype configuration, with a larger difference, only meets the "minimum" criterion (Table 2 and Table 3).

Moreover, the commercial configuration exhibits data availability ranging from 99.5% at the lowest measurement height, i.e., 40 m above the ground, to 93.0% at the highest, i.e., 200 m above the ground, with an overall vertical average availability of 98.2% (Fig. 5b). Similarly, the prototype configuration follows this trend, with data availability decreasing with altitude. The

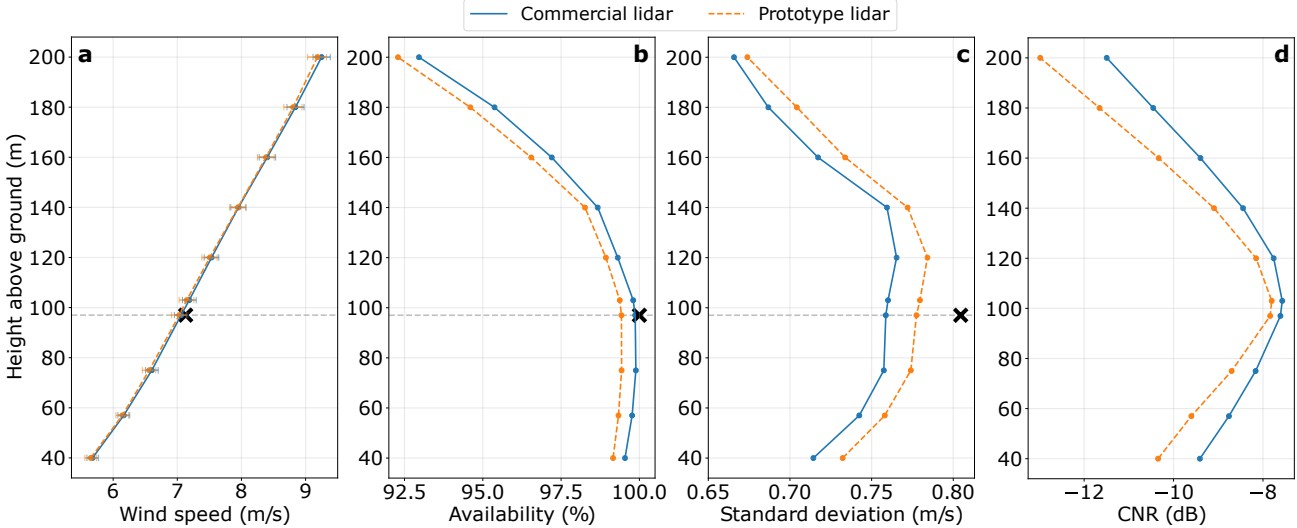

**Figure 5.** Mean vertical profiles, averaged across the 47-day dataset, of wind speed (a), data availability (b), standard deviation derived from the variance method (c), and CNR (d) measured using the commercial (solid blue curves) and prototype (dashed orange curves) configurations. In (a), error bars represent 95% confidence intervals computed via bootstrapping, illustrating the statistical uncertainty of the mean. The black crosses represent the reference measurements from the sonic anemometer, and the grey dashed vertical line marks its position at 97 m above ground.

prototype achieves a vertical average availability of 97.7%, with a minimum of 92.3% recorded at the highest measurement altitude. The prototype configuration consistently shows data availability that is, on average, 0.5% lower than the commercial configuration at nearly all measurement altitudes. Both lidar configurations exceed the 90% data availability threshold set by DNV (2009).

Fig. 6 presents the linear regression of the 30-min averaged wind speed measured by both lidar configurations in comparison to the reference instrument. Both the commercial and prototype configurations match the "best practice" criteria, with slope values of 1.0 and $R^2$ values of 0.9847 for the commercial configuration. The prototype configuration shows values that are 1% lower for the slope and similar $R^2$, but these differences are minimal and still within the acceptable range for "best practice."

**Table 3.** Acceptance criteria for KPI achievement applied on mean wind speed associated with the commercial and prototype configurations: ✓✓ denotes "best practice" and ✓ indicates "minimum" acceptance, as defined in Table 2.

| Configuration | Difference | Slope | $R^2$ | Data availability |
|---|---|---|---|---|
| Commercial | ✓✓ | ✓✓ | ✓✓ | ✓✓ |
| Prototype | ✓ | ✓✓ | ✓✓ | ✓✓ |

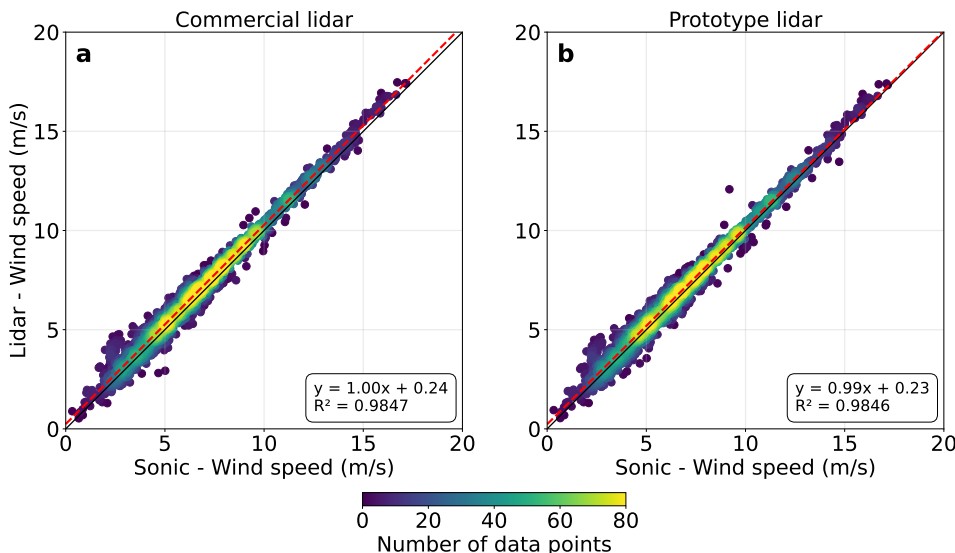

**Figure 6.** Scatter plots of the 30-min averaged wind speed measurements over the 47-day campaign, comparing the commercial lidar (a) and prototype lidar (b) with the reference sonic anemometer. Red dashed lines indicate the linear regression fits.

## 3.2 Impact of sampling rate on turbulence energy capture

The impact of increasing the sampling rate on turbulence measurement can initially be assessed using data from a sonic anemometer, specifically through the computation of along-wind velocity spectra. Integrating these spectra provides the along-wind variance, $\sigma^2$. Fig. 7 illustrates the individual spectra and the mean spectrum averaged over the 47-day dataset in both log-log and linear formats. The mean spectrum clearly follows the $f^{-5/3}$ slope, confirming the presence of the energy cascade (Fig. 7a).

The linear representation (Fig. 7b) highlights that most of the energy, associated with larger eddies, is concentrated in the frequency range from 0 to $f_{N_c} = 0.125\,\mathrm{Hz}$, corresponding to the Nyquist frequency of the LOS velocity in the commercial lidar configuration. However, additional energy, associated with smaller eddies, exists within the range from $f_{N_c}$ to $f_{N_p} = 0.5\,\mathrm{Hz}$, the latter being the Nyquist frequency of the prototype lidar configuration.

To quantify this effect, the variance was computed by integrating the spectra over two frequency ranges. First, the integration from 0 to $f_{N_c}$ simulated the variance measurable by a sonic anemometer with a sampling rate equivalent to the commercial lidar. This yielded a mean variance of $0.4712\,\mathrm{m^2/s^2}$. Second, the integration from 0 to $f_{N_p}$ simulated the variance measurable with a sampling rate equivalent to the prototype lidar, resulting in a mean variance of $0.6314\,\mathrm{m^2/s^2}$. This comparison indicates that increasing the sampling rate by a factor of 4, relative to the commercial lidar configuration, could capture an additional 34% of the variance. However, this represents the maximum possible improvement, as it is derived from measurements using a sonic anemometer, which is not affected by technical limitations such as the space-time volume averaging inside the probe of a wind lidar profiler.

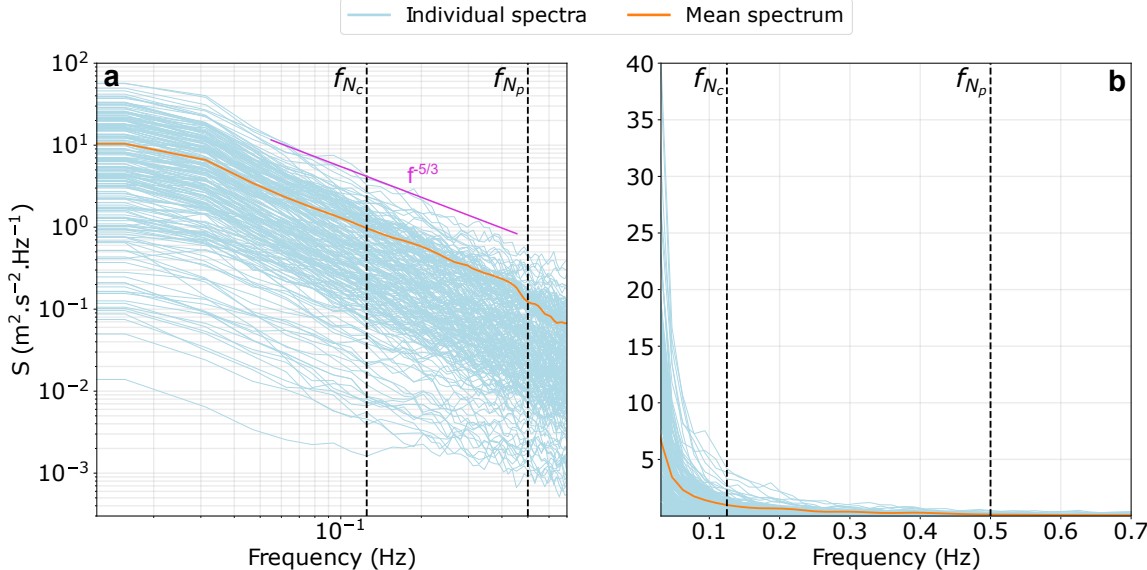

**Figure 7.** Individual spectra (light blue curves) and mean spectrum (orange curve) measured by the sonic anemometer over the 47-day measurement campaign, presented in log-log (a) and linear (b) formats. Vertical black dashed lines indicate the Nyquist frequencies, $f_{N_c}$ and $f_{N_p}$, for the commercial and prototype lidar configurations respectively. The pink solid line in panel (a) shows the classic spectral slope $f^{-5/3}$.

### 3.3   LOS velocity spectra

The determination of the instrumental noise from the spectral method involves computational fitting of the LOS velocity spectra
using a parametric expression (Eq. 2). Three weighting schemes were systematically explored to enhance fitting accuracy and minimize errors relative to the measured spectra. Fig. 8a illustrates an example of the three weighting scheme applied to a measured spectrum. This iterative process was conducted across both lidar configurations, yielding consistent results described hereafter.

The fitted spectra closely matched in the low-frequency domain, approximately up to $f = 0.1$ Hz, but strong divergences were
observed thereafter. The low frequencies weighted scheme produced a curve substantially below the measured spectra at higher frequencies, whereas the unweighted scheme yielded a curve slightly above the measured spectra in this frequency range. In contrast, the high frequencies weighted scheme provided a fit that closely matched the measured spectra across all frequencies and exhibited the lowest mean error. For instance, when applied to the prototype lidar, the mean variance was 0.2321 m$^2$/s$^2$ for all integrated fitted spectra using the high-frequency weighted scheme, compared to 0.2262 m$^2$/s$^2$ for all integrated measured
spectra. This results in an absolute error of 2.6%. Conversely, not employing any weighting during the fitting process resulted in an absolute error between the mean variance nearly three times higher, at 8.5%. Assigning weights to the low frequencies resulted in a mean absolute error exceeding six times that of the high-frequency weighted scheme, at 16.9%. Thus, the high-

frequency weighted scheme was chosen for the fitting. An example of this fitting applied to individual LOS velocity spectrum for both the commercial and prototype configurations is shown in Fig. 8b. This weighted scheme enabled the systematic

identification of the plateau at higher frequencies, characteristic of white noise. Other weighting schemes did not consistently exhibit this plateau, making it challenging to reliably determine the value of $N_i$.

## 3.4   Instrumental noise

### 3.4.1   Carrier-to-noise ratio

Fig. 5d shows that the CNR of the prototype lidar is consistently lower than that of the commercial system throughout the

altitudes of measurement, with a mean value across all heights of -8.9 dB. On average, the prototype lidar's CNR is 8.5% lower, indicating a weaker signal and thus a higher relative noise level compared to the commercial lidar. Despite this difference in magnitude, the mean vertical profiles of CNR for both lidars follow the same trend: the CNR reaches a minimum near the ground, increases to a maximum at approximately 100 m above ground, and then decreases again, reaching another minimum at the highest measurement altitude.

### 3.4.2   Comparison of the spectral and ACF methods

The spectral method yields a median variance that is 1.5 times higher than that of the ACF method for the commercial and prototype lidars (Table 4). While this suggests differences in how each method characterizes noise, the spectral method also results in a mean instrumental noise that is 30-40% lower than that of the ACF method, indicating variations in the way noise is estimated. Moreover, the spread of mean values is notably narrower when using the spectral method, particularly for the

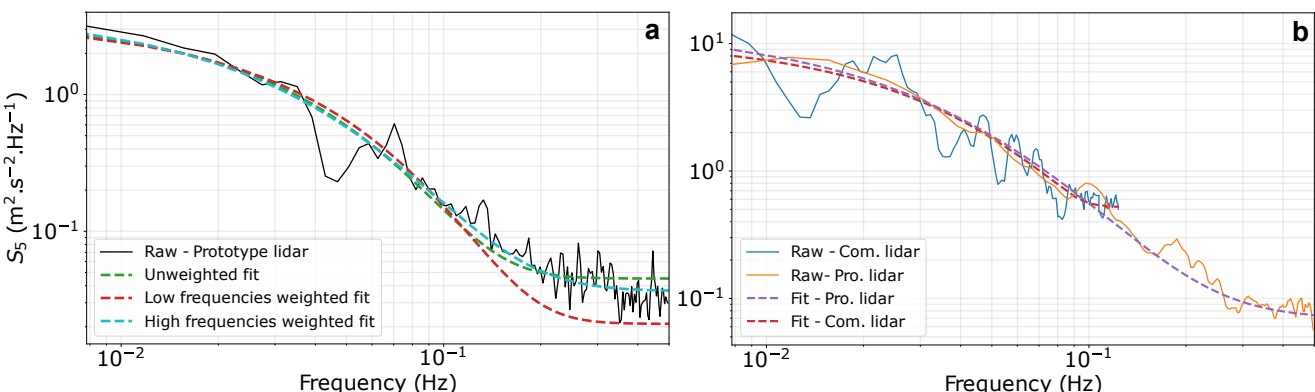

**Figure 8.** (a) LOS velocity spectrum measured by beam 5 of the prototype lidar (solid black), fitted using Eq. 2 with three different weighting schemes: unweighted (dashed green), low-frequency weighted (dashed red), and high-frequency weighted (dashed blue). This panel corresponds to the study focused on selecting the optimal weighting scheme. (b) The optimal scheme (high-frequency weighted) is applied to LOS velocity spectrum measured by beam 5 of the commercial lidar (blue) and the prototype lidar (orange).

commercial lidar, where it is reduced by half compared to the ACF method. This suggests a potential advantage in terms of consistency and stability. Given these observations, we used the spectral method to correct the measured variance, as it appeared to provide more stable estimates of instrumental noise.

To validate the applicability of the ACF method and investigate the higher spread of mean values of instrumental noise associated with this method, the ADF test was applied to each 30-min data subset. The results show that approximately 8%
of the subsets yielded p-values just above the 0.05 significance threshold, though none exceeded 0.06. This indicates that, for these subsets, the null hypothesis of non-stationarity (see Section 2.4.2) could not be rejected. Consequently, they cannot be confidently considered stationary, and the ACF method is not strictly valid for them. This limitation may partly account for the higher variability in noise estimates produced by the ACF method, as non-stationary data can lead to inconsistent results in autocorrelation-based analyses.

### 3.4.3   Contribution of instrumental noise in the measured LOS velocity variances

The parametric expression (Eq. 2) used to fit the LOS velocity spectra measured by beam $i$ enables the identification of the power spectral density of instrumental noise, $N_i$, and the derivation of the variances, $\sigma^2_{n_i}$ (Eq. 4). Fig. 9 compares the mean magnitude of $\sigma^2_{n_i}$ to the mean variance of the net contribution from atmospheric turbulence, $\sigma^2_{p_i}$, corrected for instrumental noise at scales observable by the commercial and prototype lidar profilers.

The mean values of $\sigma^2_{n_i}$, which are nearly identical across all beams, were found to be 0.0108 m$^2$/s$^2$ for the commercial configuration (Table 4). A similar trend was observed for the prototype configuration, although the mean variance of instrumental noise was 68% higher, at 0.0181 m$^2$/s$^2$ (Table 4). Notably, the contribution of instrumental noise variance to the total variance, $\sigma^2_{b_i}$ (Eq. 3), was found to be 4.8% and 7.4% for the commercial and prototype lidar configurations, respectively.

The mean variances, $\sigma^2_{p_i}$ were consistently higher for measurements obtained with the prototype configuration. Across all
beams, the mean value was 0.2288 m$^2$/s$^2$, which is 7.8% higher than the corresponding mean value for the commercial lidar measurements.

### 3.5   Along-wind standard deviation

Fig. 10 presents scatter plots of the along-wind standard deviation, $\hat{\sigma}$, derived from the variance method applied on measurements of both lidar configurations compared to the standard deviation, $\sigma$, obtained from the reference sonic anemometer. The

**Table 4.** Median and mean ($\pm$ spread) variance of instrumental noise for commercial and prototype lidars, computed from the LOS velocity measurements across all beams using spectral and ACF methods.

| Methods | Commercial lidar | | Prototype lidar | |
|---|---|---|---|---|
| | Spectral | ACF | Spectral | ACF |
| Median (m$^2$/s$^2$) | 0.0076 | 0.0050 | 0.0129 | 0.0081 |
| Mean $\pm$ spread (m$^2$/s$^2$) | 0.0108 $\pm$ 0.0102 | 0.0148 $\pm$ 0.0228 | 0.0181 $\pm$ 0.0175 | 0.0237 $\pm$ 0.0294 |

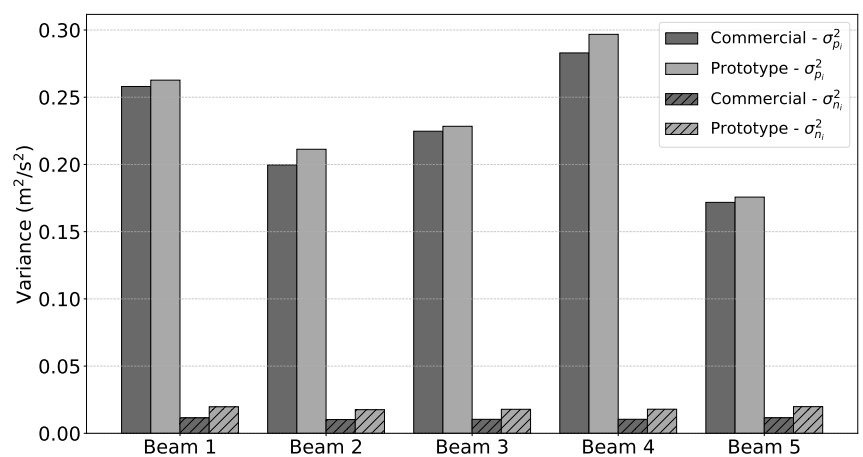

**Figure 9.** Mean variance of the net contribution from atmospheric turbulence ($\sigma_{p_i}^2$), corrected for instrumental noise derived from the spectral method, measured by each beam $i$ at scales observable by the commercial (dark gray) and prototype (light gray) lidar profilers. Dashed areas represent the mean variance of instrumental noise, $\sigma_{n_i}^2$. The averages were computed over the 47-day dataset.

**Table 5.** Error Statistics of the along-wind standard deviation derived from the variance method, corrected for instrumental noise, applied on measurements collected by the commercial and prototype lidars in comparison to the reference sonic anemometer.

|  | Bias (m/s) | MAE (m/s) | RMSE (m/s) | $R^2$ | Relative Error (%) |
|---|---|---|---|---|---|
| Commercial lidar | -0.0639 | 0.0886 | 0.1218 | 0.9138 | 7.8 |
| Prototype lidar | -0.0466 | 0.0678 | 0.0871 | 0.9574 | 5.7 |

prototype configuration demonstrates superior performance across all error metrics, with bias, MAE, and RMSE approximately 25% lower than those of the commercial configuration (Table 5). Additionally, the coefficient of determination is 5% higher. There is also a reduction in the relative error of the mean standard deviation, with the prototype configuration showing values of 5.7% compared to 7.8% for the commercial configuration.

Fig. 11 presents wind speed-binned estimates of $\hat{\sigma}$ compared to estimates of $\sigma$ (black curve) as a function of binned-averaged wind speed. For all wind speeds, the standard deviation measured by the sonic anemometer consistently remains higher than that derived from both lidar configurations. Below wind speed of 8 m/s, the standard deviation values from both lidar configurations closely match each other. Within this wind speed range, the standard deviation associated with the commercial lidar is 2.7% higher than that from the prototype configuration. However, above this wind speed threshold, the standard deviation associated with the prototype configuration increases more rapidly with wind speed compared to the commercial lidar. In this wind speed range, the standard deviation associated with the prototype lidar is 13.0% higher than that associated with the commercial configuration. For all wind speed ranges, the prototype lidar measurements exhibited a mean standard deviation and variance that are 3.5% and 7.2% higher, respectively, than those of the commercial configuration.

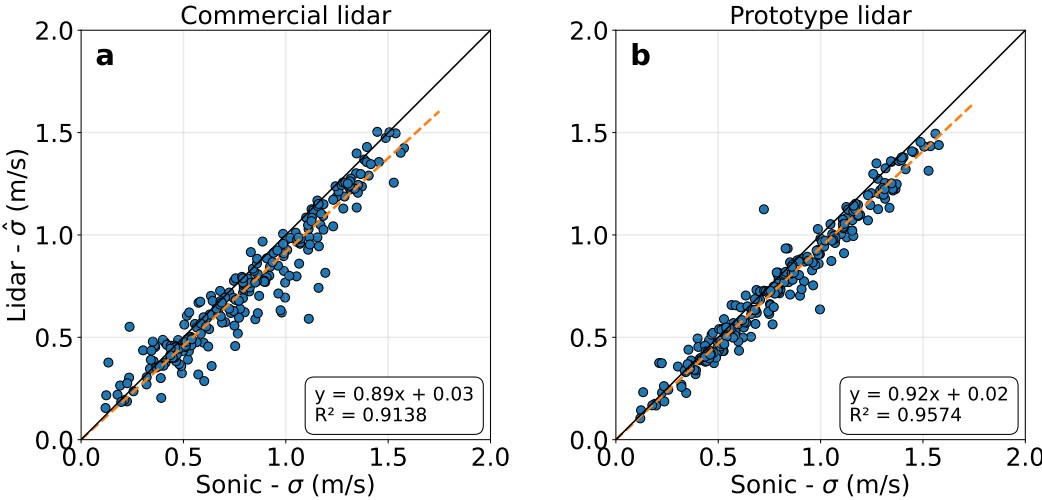

**Figure 10.** Scatter plots of along-wind standard deviation, $\hat{\sigma}$, derived from the variance method applied on measurements of the commercial and prototype lidar configurations versus standard deviation, $\sigma$, derived from the reference sonic anemometer. The standard deviation estimates are restricted to cases where wind direction was aligned with one pair of opposite beams.

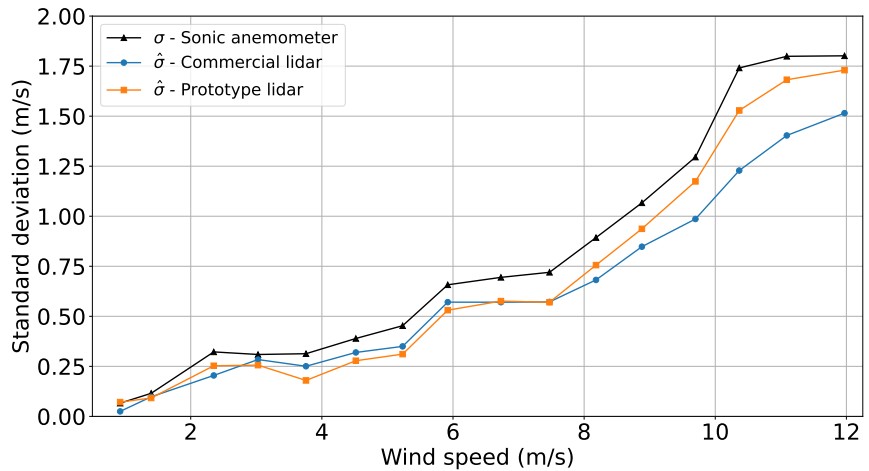

**Figure 11.** Along-wind standard deviation, $\hat{\sigma}$, derived from measurements of the commercial lidar (blue curve) and prototype lidar (orange curve), compared to the standard deviation, $\sigma$, obtained from reference sonic anemometer measurements (black curve) as a function of wind speed.

## 4  Discussion

The increased sampling rate resulted in a relatively slight 0.5% reduction in data availability compared to the commercial configuration over the 47-day dataset. While this difference is minimal, longer measurement campaigns, typically lasting over

a year for wind site characterization, may accumulate more instances of data loss due to environmental factors, hardware limitations, or maintenance events, potentially making the impact of reduced availability more noticeable over time. Following the measurement campaign presented in this paper, the prototype configuration was installed in December 2022 on Planier Island in the Mediterranean Sea, where it remains operational. The wind characteristics derived from the full year of 2023 are presented in Thiébaut et al. (2024), including a detailed analysis of data availability. Encouragingly, up to 160 m above sea level, annual data availability exceeded the 90% threshold considered best practice. Beyond this height, availability gradually declined, reaching below 70% at 220 m. While this highlights an area for further optimization, the prototype lidar has already demonstrated strong performance at critical measurement heights.

Moreover, the prototype configuration performed comparably to the commercial setup in terms of mean wind characteristics. While the commercial configuration met the "best practice" threshold for all key performance indicators (KPIs), the prototype also achieved this standard, with the exception of mean wind speed differences, where it met the "minimum acceptance" level within the best practice range. This result is promising, as it confirms that the prototype lidar meets industry standards while offering opportunities for further refinement.

Reducing the accumulation time increases the sampling rate, which helps limit temporal averaging of the wind signal and preserves more of the high-frequency variance within the instrument's resolvable range. However, this also requires a careful evaluation of instrumental noise and its associated variance to ensure that observed changes in variance can be confidently attributed to atmospheric turbulence rather than measurement artifacts. In this study, the noise-induced variance was estimated using two independent methods. For both lidar configurations, the noise levels were consistent with values reported in previous studies, such as the WindCube lidar analysis by Mann et al. (2009), supporting the reliability of our estimates. As expected, the higher sampling rate in the prototype lidar led to increased instrumental noise due to the reduced number of transmitted pulses (Pearson et al., 2009). The noise variance constituted approximately 5% of the total variance for the commercial configuration and over 7% for the prototype. While non-negligible, this contribution was accounted for in all variance-based metrics.

However, the impact of increased sampling must be interpreted with care. If the sampling frequency is too low relative to the turbulent fluctuations present in the flow, aliasing occurs: unresolved high-frequency energy is folded into lower frequency bands, distorting the spectral distribution. While the reduced accumulation time increases the sampling frequency and helps mitigate aliasing effects, it does not fully recover the true spectral shape, given the finite temporal resolution. Therefore, the observed increase in variance with the prototype lidar should primarily be attributed to reduced temporal and spatial filtering, rather than a direct "gain" in turbulent energy capture. This distinction is critical.

Increasing the sampling rate does not linearly "capture more variance" from smaller eddies. Instead, the accumulation time acts as a low-pass filter on the LOS velocity signal, attenuating contributions from high-frequency turbulent fluctuations. Systems with longer accumulation times, such as the 0.8 s used in the commercial lidar, are more affected by this filtering, especially at higher wind speeds where the effective probe length, $L_{\text{eff}}$, becomes longer due to advection. The effective probe length refers to the spatial distance over which the LOS wind velocity is effectively averaged, accounting for both the fixed probe length and the distance traveled by air during the accumulation period. It can be estimated as:

$$L_{\text{eff}} = L_{\text{probe}} + U\Delta t, \tag{11}$$

A longer effective probe length increases spatial averaging and attenuates variability in the measurements. In contrast, the prototype lidar, with a shorter 0.2 s accumulation time, experiences less temporal averaging and thus preserves a greater portion of the wind variance from smaller-scale motions. This contributes to the higher along-wind variances observed in the prototype configuration compared to the commercial system. Despite this, the observed increase in variance remains significantly below the theoretical benefit expected from increasing the LOS sampling rate, as determined through sonic anemometer measurements. The measurement volume of a sonic anemometer is effectively point-like, in comparison to the much larger probe length of wind lidar profilers. The anemometer is essentially free of the probe-time averaging effect, which enables it to capture the wind signature of very small eddies.

## 5 Conclusions

The prototype configuration of the WindCube v2.1 lidar profiler demonstrated comparable performance to the commercial system in terms of mean wind characteristics and data availability, meeting industry standards. Moreover, this study demonstrates that increasing the sampling rate of LOS wind velocity measurements by reducing the accumulation time in pulsed wind lidar systems effectively mitigates temporal and spatial averaging effects inside the probe volume, thereby improving turbulence measurements. The prototype configuration, with a fourfold increase in sampling rate compared to the commercial system, preserved additional variance by reducing both spatial and temporal averaging, rather than directly resolving smaller eddies, particularly at higher wind speeds, where advection lengthens the effective probe length. Despite these gains, the increased sampling rate introduced trade-offs, including higher instrumental noise and a slight reduction in data availability. The noise was systematically corrected, and its impact was found to be manageable.

The findings underscore the importance of carefully balancing temporal resolution, noise, and probe length when configuring lidar systems for turbulence retrieval. While commercial lidars can be programmed by the manufacturer to match the prototype's sampling rate, it is advisable to validate the lidar's performance through certification after such adjustments. Looking forward, optimizing both accumulation time and probe length in tandem may enhance the ability of wind lidar systems to capture turbulence more accurately. However, such changes must be approached cautiously, as reducing these parameters can also compromise data availability, especially under challenging atmospheric conditions. Balancing resolution and reliability will be key to supporting broader applications in wind energy assessments and atmospheric boundary layer research.

## Author contributions

MT identified the problematic, performed the analysis and drafted the paper. LM performed the mathematical demonstration of Eq. 1, available in the supplementary material, and provided guidance on the manuscript structure and its review. FD and FG reviewed the manuscript.

**Data and code availability**

The data is owned by a private consortium with proprietary rights and confidentiality obligations, precluding its sharing alongside this paper.

**Acknowledgments**

We would like to acknowledge the team at Vaisala, including Mathias Régnier, Loïc Mahe and Cristina Benzo, for their support in providing and configuring the prototype lidar.

**Competing interest**

The authors declare that they have no conflict of interest.

**Financial support**

This work was made possible through the support of France Energies Marines and the French government, managed by the Agence Nationale de la Recherche under the Investissements d'Avenir program, with the reference ANR-10-IEED-0006-34. This work was carried out in the framework of the POWSEIDOM project.

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
