# Peer review of "Evaluating enhanced sampling rate for turbulence measurement with wind lidar profiler"

_Wind Energy Science, 2024_

## Referee Comment (RC2)

**Referee's comments to wes-2024-93**

**General comments**

This paper addresses lidar improvements with a neat comparison of two new lidar prototypes with commercial systems. The finding that a reduced sampling rate is the best improvement is however poorly supported by the data.

The main drawback of the work is the lack of reference turbulent quantities to compared with. One of the systems was deployed close to a sonic anemometer but this valuable instrument is deliberately omitted. The basis on which improved turbulence estimates are claimed are mainly two and not convincing:

- Increased variance with respect to the reference lidar is by itself not indicative of improvement. As also mentioned in the introduction, lidars can overestimate variances due to cross-contamination, so how do we know that the increased sampling rate is not indeed exacerbating a positive bias in the variance? Increased variance could also come from noise, and this is has not been ruled out either.
- The reduced noise estimated from the spectra of $w$ is also not compelling. Increasing sampling rate extends the spectrum to higher frequency (Fig. 9), so the behavior of the fitting can change significantly. It is also mentioned that for the commercia lidar a noise plateau was not identified, so we cannot trust noise estimates from the reference lidar so what observed in Fig. 10a can be a numerical artifact

Also, the spectral analysis shows that spectra are very noisy and therefore the results should be interpreted more carefully. For instance, the laminar flow case in Fig. 11 is very questionable as laminar flow generally does not occur in the field and also because the supposedly laminar spectrum has more variance than the turbulent spectrum.

It is suggested to profoundly revise this work to make the most out of this useful dataset:

1. Calculate the turbulent statistics form the sonic (or even cups) as well and use it as reference
2. Do not provide overall biases only, but also RMS error on a 10-minute basis or, even better, scatter plot like the one in Fig. 5 for lidar vs sonic
3. For the lidar with reduced probe volume where there is no met mast and very few data points, consider a smaller section with a lot of caution advised in the interpretation of the results
4. Evaluate lidar noise also using a non-spectral approach, like the autocorrelation method by Lenschow et al., 2000 ([https://doi.org/10.1175/1520-0426(2000)017<1330:MSTFOM>2.0.CO;2](https://doi.org/10.1175/1520-0426(2000)017<1330:MSTFOM>2.0.CO;2))
5. The introduction could mention the effect of pulses accumulation, which is different from the sampling rate. The accumulation acts as a low-pass filter in the time domain in an analogous way as the probe average does in the spatial one. The sampling rate refers more to how quickly the lidar moves through the scan cycle, regardless of how long it takes to measure a single LOS.

These are some modifications that would bring the paper to the standards of the other publications in the topic.

**Specific comments**

L71: "mea" instead of "mean"

L77: is the increased sampling rate achieved through a faster accumulation or a higher pulse repetition frequency? In the second case, the maximum range may be reduced, and it should be explained.

L81: sampling rates of 0.25 Hz for wind speed may be misleading. The lidar uses a moving averaging window of 5 beams, so it does deliver a new wind speed estimate every second, but these estimates are not independent. This time overlapping effect should be made clear.

L 94: please explain what the test requirements were to consider it as "passed".

L130: the explanation of the rotation of velocity is unclear. In general, $V_x$ and $V_y$ are not 0, but after rotation $v = 0$ (not $V_y$ as indicated). Aligning the x axis to North is also not the common practice in atmospheric science, where x is W-E and y is S-N, and it may be worth mentioning this as well. Please add Fig. 1 angles and axis clearly indicated for readers that are unfamiliar with this technique

Equations 1 and 2: $b$ terms that should be the LOS velocities are not defined.

L201: it is true that the inertial subrange is limited to the right by the viscous regime where dissipation reduces TKE, but it is also limited to the left by the integral scales that supply TKE, please add this detail.

Eq 9: the | symbol to indicate the range of frequencies may be mistaken for an integration. If a fit is instead performed in this region, it would be better to remove it and explain that it is a fitting operation in the inertial subrange.

L245: is the specification of 1% relative to the error over 10 minutes or the whole dataset? Please specify.

L255: have you considered that the increased difference close to the ground may be due to the lidar with reduced probe length being able to resolve better nonlinear mean wind shear?

L272: the increase in interquartile range cannot be automatically ascribed to a better sensitivity since it could very much be noise (instrumental or statistical). The fact that larger increases in standard deviation are seen at high altitude is also suspect in this sense, since one could expect the reduced probe length to lead to more recovery of turbulence variance close to the ground where length scales are smaller. If it happens at larger range, it could be noise not sensitivity.

Fig. 5: please add the colorbar of data density.

Fig. 6: please make the box and whisker format consistent between the two subplots.

L283: "iterative" may not be the right word, "trial and error" maybe?

L330: it is confusing saying that $\beta = 5/3$ was imposed for the dissipation energy, but then $\beta < 1$ were excluded. Is this a two-step process where first we fit $\beta$ to the whole spectrum, then if it passes the check it is used for the dissipation energy with a new fit in the inertial subrange and $\beta = 5/3$? Please clarify.

L359: the integral length scale is not associated with a peak in the spectrum (not premultiplied), but it is by definition its value at 0 frequency, as shown in Pope 2020, Eq. 3.114. Please remove or rephrase.

---

## Author Comment (AC1)

The updated version of the manuscript now focuses exclusively on the increased sampling rate, for which a 47-day dataset was collected. This dataset is accompanied by reference turbulence measurements provided by a sonic anemometer installed on a nearby met mast. The lack of reference data in the original manuscript was another concern raised by you and the second reviewers. In this revised version, we incorporate the sonic anemometer dataset for more robust comparisons.

The paper now addresses the impact of the increased sampling rate on data availability and key performance indicators (KPIs), such as the slope of the scatter plot between the mean wind speed measured by the lidar and that measured by the reference, the correlation coefficient ($R^2$) of the linear regression of the scatter plot, and the mean absolute difference in mean wind speed.

We then narrowed the focus to examine the effect of the increased sampling rate on the variance and standard deviation measured by both lidars, as these values are used to compute turbulence intensity (TI), the most used metric in the wind power industry to quantify turbulence. In the first version of the manuscript, comparing multiple turbulence metrics (such as dissipation rate and integral length scale) created confusion, so we chose to concentrate on variance and standard deviation. These estimates were corrected for the variance of instrumental noise, which was quantified using two methods: a spectral approach and an autocorrelation approach, as suggested by the second reviewer.

We estimate that the updated version of the manuscript is approximately 90% revised compared to the original version.

**General comments:**

In the manuscript, two modified prototype versions of the WindCube vertical profiling pulse wind lidar are compared to the default WindCube v2.1. The authors look at one version featuring four times faster sampling rates with accordingly shorter accumulation times and a second version with reduced pulse length resulting in shorter probe volumes. They assess the impact of the modifications on data availability, mean wind speed, noise level, standard deviations of turbulent velocity fluctuations, integral length scale of turbulence, velocity spectra and dissipation rate.

The study is of high importance for the field because profiling wind lidar suffers from their limited ability to measure turbulence accurately. Therefore, every effort to increase this ability is sought after. The experimental setup, modifications to the lidar units, and methods for assessement of the results are overall well described.

In the overall acceptable introduction and the good and well-written description of the methods, Thiebaut et al. should be more accurate in some of their claims (see specific comments).

We have revised the introduction, which, as you pointed out, was previously acceptable but could benefit from more clarity and precision.

- The experiment with the WindCube with reduced probe length is only 4 days long. This very short trial length minimizes the value of the conclusions drawn from it and is a major drawback for the study.

We acknowledge that the short duration of the experiment with the WindCube and the reduced probe length (only 4 days) is a significant limitation of the manuscript, as pointed out by you and the second reviewer as well. In response, we have removed the analysis of the reduced probe length in the updated version of the manuscript. This is an ongoing work that is not ready to be published.

- The manuscript would benefit from a more thorough explanation of the theory behind the tested modifications. For example, it remains unclear how the increased sampling rate of the

modified lidar leads to a reduction of noise, although every LoS measurement will have a higher potential for noise if accumulation times are reduced.

We acknowledge this mistake and agree that our initial comparison of the power spectral density of noise was not appropriate. The correct approach is to compare the variance of noise, which is obtained by multiplying the power spectral density by the Nyquist frequency. After correcting this, we found that the variance of noise associated with the prototype configuration is 68% higher than that of the commercial configuration. The increased sampling rate leads to higher instrumental noise compared to the commercial configuration, as the variance of noise is inversely proportional to the number of transmitted pulses, as noted by Pearson et al. (2008). Now, our results align with this expectation. We have revised the manuscript to address this point.

- The conclusions and discussion section would benefit from some more reflection on theoretical considerations and practical implications.

The conclusions and discussion but also the introduction and section 2.1 (presenting the prototype configuration) have been revised to include additional reflections on theoretical considerations and practical implications, providing a more comprehensive discussion of the findings.

*Specific comments:*

- l. 26: Considering a beam diameter of 3cm, only a probe length of around 17km fills up a probe volume 12 cubic meters. Thus, the statement of lidars providing averages of "up to several dozen cubic meters" appears to be a vast exaggeration. Such a statement should be backed up by references.

You are right. We have reformulated the sentence: "Anemometers estimate wind speed over a small volume of just a few cubic centimeters, whereas pulsed wind lidar profilers provide an average over a cylindrical probe several dozen meters long with a cross-sectional diameter of less than 1 cm (Fig. 1)". Lines 31-33, page 2.

No formal reference is provided for the dimensions, as they were supplied by the manufacturer during the collaboration for this work.

- l. 38: The ZX 300, one of the two market-leading wind profilers, requires one second to complete a full scanning circle. The statement that "Lidar profilers require several seconds to complete a full scanning circle" is therefore wrong and must be corrected.

We have added the word 'Pulsed' at the beginning of the sentence to exclude the ZX lidar, which uses continuous wave technology. Line 58, page 3.

- l. 80: "the WindCube lidar collects data at each location in 1 second" is wrong because it samples 5 beams within 4 seconds. So, on average the data accumulation time per beam direction cannot be higher than 0.8 seconds. And in practice some dead time for swinging the beam must be accounted for.

We are right. We don't mention anymore the sampling rate of wind speed since we are now only addressing statistics performed on LOS velocities. Here is the new version of this statement: "In its standard commercial configuration, the WindCube lidar collects data at each position for approximately 0.8 seconds before transitioning to the next. Including the time required to shift between positions, a full DBS scan is completed in 4 seconds. This corresponds to a LOS velocity sampling rate of 0.25 Hz." Lines 95-97, page 4.

- l. 89: The authors should not report an "improvement" in the method section. Instead they can write e.g., about the "modification".

We changed "improvement" by "modification". Line 83, page 4.

- l. 89: Accumulation time is reduced by 70% to which value (0.3*0.8s=0.24s)? In line 88 1Hz and 4Hz, although there are five beams to be sampled (1Hz*5=>0.2s per beam). So, either the 4Hz or the 70% reduction are wrong.

The 4 Hz value was incorrect, but it is no longer mentioned since we now compute the standard deviation and variance only on LOS velocities, not on wind speed. Also, we removed the "70%" value since it requires confidential information to explain this specific percentage.

- l. 93: Please describe why you chose 30-min intervals, instead of 10-minutes commonly used in wind industry.

It is a common remark. A 30-minute temporal window was chosen for subsequent analysis, deviating from the standard 10-minute window typically used in the wind energy industry. This decision was guided by the need to address random errors in turbulence measurements, as highlighted by Lenschow et al. (1994). For a given averaging time T, there is a systematic difference between the true flux and the ensemble average of the time means of the same quantities, known as systematic error. This error decreases as T increases. Additionally, the error variance - representing the random scatter of individual realizations - also diminishes with longer T. Therefore, the use of a 30-minute averaging period, common in atmospheric science, is justified over the 10-minute window.

We have added this text: "The choice of a 30-minute window deviating from the standard 10-minute window typically used in the wind energy industry was informed by considerations of reduction of random errors in turbulence measurements, as discussed by (Lenschow et al., 1994). Lines 144-146, page 7.

- l 111: The short trial duration of 4 days is a major drawback of the study. Please explain why it was not possible to perform a longer experiment and how the short trial durations impeded the study.

We removed this analysis. It was just a preliminary investigation.

- 2.1.1 & 2.1.2: Please elaborate on the effect of both modifications from a theoretical standpoint. Why did you choose these modifications. What is the trade-off between duty cycle, accumulation time and sampling rate? What are the technological limitations? Why is the default configuration of the WindCube different?

Now a full section is dedicated to this (section 2.1, page 4-5).

- l. 129: Mean velocity does not have a standard deviation, but the fluctuations superimposed on the mean velocity and there is no mean velocity across the wind propagation. Please reformulate.

It is a mistake. It should have been "the standard deviation (to) the mean" and not "(of) the mean". Anyway, we removed this sentence.

- l. 242: Please name the first and last measurement height (40m and 200m) to help the reader. Please consider to show how the 0.5% reduction is distributed along the vertical profile.

The sentence is not modified: "Moreover, the commercial configuration exhibits data availability ranging from 99.5% at the lowest measurement height, i.e., 40 m above the ground, to 93.0% at the highest, i.e., 200 m above the ground, with an overall vertical average availability of 98.2% (Fig. 4b).". Lines 278-280, page 11.

Moreover, we have added Fig. 4b to show the vertical variation of data availability.

- ll. 244-249: IEC (2017) does not prescribe any "Best Practice" criteria. Where do the 1% and 1.5% thresholds come from?

This is a misunderstanding. The term 'best practice' refers to KPIs defined by DNV-GL for certifying lidar profilers against met masts for mean wind speed measurements. We used these KPIs to evaluate the ability of both the commercial and prototype lidar to measure mean wind speed. Please refer to Sections 2.6 (page 10) and 3.1 (page 11), as well as Tables 1 and 2, for further details.

- Fig. 8: Consider the add a titles to the figure.

The figure already includes a title in the caption, ensuring clarity without redundancy.

- Fig. 9: The legend entries seem to be wrong. "Fit - Com. lidar" ranges all the way to the highest frequency.

Thank you. It was wrong. We corrected it (Fig. 7b).

- Fig. 9: Consider to merge the two sub figures into one, since only the fitted line is different and could be compared more easily in one plot.

We did it. See Fig. 7b.

- Fig. 9: The distribution of frequency bins is different for the com. and the pro. lidar. Consider to create logarithmically spaced frequency bins for both lidars and average the spectral energy within these bins. This would make the comparison easier.

We have implemented logarithmically spaced frequency bins for both the commercial and prototype lidars and averaged the spectral energy within these bins. This adjustment simplifies the comparison between the two configurations. See Fig. 7b.

- 3.4: The noise assessment would benefit from a figure.

We have added a table (Table 3) and a figure (Fig. 8) to enhance the noise assessment, providing a clearer and more comprehensive presentation of the data.

- ll. 342-347: 0.5% reduction in data availability are not a "slight" reduction. And 1-2% deviation in mean wind speed are significant and in the order of magnitude of the total measurement uncertainty of profiling wind lidars. The authors clarify that a deviation in wind speed beyond 1.5% prohibits certification (l. 248). Here they call it "very slight bias". I understand that the manuscript focuses on turbulence estimates, still the deviations in mean wind speed must be analyzed equally carefully.

We acknowledge the reviewer's comment regarding the 0.5% reduction in data availability and the 1-2% deviation in mean wind speed. The manuscript does indeed focus primarily on turbulence estimates, but we recognize the importance of addressing these deviations carefully. The 0.5% lower data availability in the 47-day dataset is noted, and we emphasize that over a longer campaign (typically lasting over a year), this difference may be more significant. Regarding the deviation in mean

wind speed, it is now assessed using key performance indicators (KPIs), which are introduced in Section 2.6 (page 10) and further discussed in Section 3.1 (page 11). We have clarified the analysis of mean wind speed performance throughout the manuscript to ensure its significance is appropriately addressed.

- l. 350: The WindCube with increased sampling rate has a probe length of 23m that effectively acts as a low-pass filter. Still, the authors assume that an increase in sampling rate from 0.25Hz to 1Hz per beam leads to a "greater sensitivity to smaller-scale fluctuations". With a mean wind speed of e.g. 7m/s the sampling rates correspond to eddy sizes of 28m and 7m respectively. The authors have to explain why they still think, that the higher standard deviation for all three turbulent velocity components can be caused by the beneficial aspects of an increased sampling rate.

We have observed that the along-wind variance, after noise correction, is approximately 7% higher for the prototype configuration compared to the commercial configuration. This suggests that the prototype's increased sampling rate may allow it to capture the energy of smaller eddies more effectively. However, the observed improvement falls short of the expected increase, as indicated by the sonic anemometer data, which suggests that a higher sampling rate could capture 34% more energy (see Section 3.2). Additionally, as shown in Fig. 10, this effect is anticipated to be more pronounced at higher wind speeds, such as 15 m/s.

- l. 399: As commented before, the effect of increased sampling rate on mean wind speed appears significant and also the impact on data availability seems to be stronger than "slight".

We have added this:

"The increased sampling rate resulted in a relatively slight 0.5% reduction in data availability compared to the commercial configuration over the 47-day dataset. While this difference is minimal, it may become more noticeable over longer measurement campaigns, which typically last over a year for wind site characterization. Following the measurement campaign presented in this paper, the prototype configuration was installed in December 2022 on Planier Island in the Mediterranean Sea, where it remains operational. The wind characteristics derived from the full year of 2023 are presented in Thiébaut et al., (2024), including a detailed analysis of data availability. Encouragingly, up to 160m above sea level, annual data availability exceeded the 90% threshold considered best practice. Beyond this height, availability gradually declined, reaching below 70% at 220m. While this highlights an area for further optimization, the prototype lidar has already demonstrated strong performance at critical measurement heights.". Lines 408-416, page 19.

*Technical corrections:*

*All the technical corrections have been addressed. Some of the changes may not be immediately apparent, as the text related to certain issues has been removed.*

l. 71: "the mea(n) wind speed"

l. 147: "pair of (parallel) beams"

l. 167: "each...subset( )"

l. 259: "illustrated (further / in the form of scatter plots) in Fig. 4."

l. 274: "the vertical (profile) of the mean..."

l. 276: "Notably, the (deviations) were..."

l. 283: "scheme(s)"

l. 193: "three (times) higher"

l. 331: "configuration (with) increased..."

l. 334: "not exceed(ing) 5.1m/s"

l. 381: "potential(ly) improved"

**Not addressed anymore in the updated version of the manuscript.**

The analysis associated with this text has been removed, and consequently, the corresponding section has also been omitted from the manuscript.

l. 125: Please mention the special role of the vertical beam for this study. Many of the results shown are based on the fifth beam only.

l. 176: Noise is not(!) due to relative motion between the source and the observer, otherwise it would be the signal. Please rewrite.

Fig. 6: Please rearrange the box plots to maintain the u,v,w/b5 order.

l. 239: The absence of reduction in 100% data availability appears to be insignificant and conclusions on the effect of reduced probe length cannot be drawn from the available data. A four day period with 100% availability is not representative for a commercial measurement campaign in which the data availability of the commercial lidar is <100%. This must be reflected on in the text. What are your expectations from theory? Does the increased sampling rate reduce the duty cycle, so that there is less total measurement time?

l. 253: How does the relative deviation of 1.3% between prototype and commercial unit relate to typical measurement deviations between two random commercial lidar units? Can the results be attributed to the modifications with certainty?

ll. 266-268: Slopes below unity and positive intercepts are always expected for standard linear regression with randomly scattered data. Consider using Deming regression to assume identical random errors for the prototype and commercial lidar.

l. 295: Why are the spectral plots for reduced probe length not shown? Same for l. 305.

Fig. 11: It is unclear why the "Laminar flow" curve is considered laminar flow although it contains the highest total spectral energy of all four examples. Where would laminar flow come from in field experiments?

l. 109: It is unclear why a "50% reduction in pulse duration" leads "to a reduction in the probe length from 23m to 15m". Due to the linear relationship, the reader would expect that a 33% reduction in pulse duration leads to this reduction in probe length. Please clarify.

The probe length is not addressed anymore.

**References**

Lenschow, D. H., Mann, J., & Kristensen, L. (1994). How long is long enough when measuring fluxes and other turbulence statistics?. *Journal of Atmospheric and Oceanic Technology*, *11*(3), 661-673.

Pearson, G., Davies, F., and Collier, C.: An analysis of the performance of the UFAM pulsed Doppler lidar for observing the boundary layer, Journal of Atmospheric and Oceanic Technology, 26, 240–250, 2009.

---

## Author Comment (AC2)

**General comments**

This paper addresses lidar improvements with a neat comparison of two new lidar prototypes with commercial systems. The finding that a reduced sampling rate is the best improvement is however poorly supported by the data.

The main drawback of the work is the lack of reference turbulent quantities to compared with. One of the systems was deployed close to a sonic anemometer but this valuable instrument is deliberately omitted. The basis on which improved turbulence estimates are claimed are mainly two and not convincing:

Increased variance with respect to the reference lidar is by itself not indicative of improvement. As also mentioned in the introduction, lidars can overestimate variances due to cross-contamination, so how do we know that the increased sampling rate is not indeed exacerbating a positive bias in the variance? Increased variance could also come from noise, and this is has not been ruled out either.

Exactly. Lidar can either overestimate or underestimate the variance of along-wind velocity ($u$) when the variance is computed from the reconstructed velocities provided by the lidar. This is known as the cross-contamination effect. To mitigate this, we focused our analysis on cases where the wind is aligned with one pair of opposite beams (beam 1/beam 3 or beam 2/beam 4) and computed the along-wind variance by combining the variance of the LOS velocities from these beams. In the specific case where the wind is aligned with the pair of beams 1 and 3, we have $\sigma_x^2 = \sigma_u^2$. Conversely, when the wind is aligned with the pair of beams 2 and 4, it holds that $\sigma_y^2 = \sigma_u^2$. Also, under these conditions, it can be reasonably hypothesized that the covariance term, $\sigma_{uv}$ is negligible (e.g., Newman et al., 2016), where $v$ represents the cross-wind velocity.

We have also corrected the variance for noise, which was quantified using two approaches: a spectral approach and the autocorrelation approach that you suggested.

The reduced noise estimated from the spectra of $w$ is also not compelling. Increasing sampling rate extends the spectrum to higher frequency (Fig. 9), so the behavior of the fitting can change significantly. It is also mentioned that for the commercia lidar a noise plateau was not identified, so we cannot trust noise estimates from the reference lidar so what observed in Fig. 10a can be a numerical artifact

Also, the spectral analysis shows that spectra are very noisy and therefore the results should be interpreted more carefully. For instance, the laminar flow case in Fig. 11 is very questionable as laminar flow generally does not occur in the field and also because the supposedly laminar spectrum has more variance than the turbulent spectrum.

It is suggested to profoundly revise this work to make the most out of this useful dataset:

Thank you very much for your thorough review and valuable feedback. Initially, the study focused on evaluating the impact of two modifications to the lidar system: (1) reducing the probe length and (2) increasing the sampling rate of the WindCube v2.1 lidar profiler. These modifications were assessed separately for their influence on turbulence measurements. The updated version of the manuscript now focuses exclusively on the increased sampling rate, for which a 47-day dataset was collected. This dataset is accompanied by reference turbulence measurements provided by a sonic anemometer installed on a nearby met mast. The lack of reference data in the original manuscript was another concern raised by the reviewer. In this revised version, we incorporate the sonic anemometer dataset for more robust comparisons.

In the updated version of the manuscript, we narrowed the focus to examine the effect of the increased sampling rate on the variance and standard deviation measured by both lidars, as these values are used to compute turbulence intensity (TI), the most commonly used metric in the wind power industry to quantify turbulence. In the first version of the manuscript, comparing multiple turbulence metrics (such as dissipation rate and integral length scale) created confusion, so we chose to concentrate on variance and standard deviation. These estimates were corrected for the variance of instrumental noise, which was quantified using two methods: a spectral approach and an autocorrelation approach.

We estimate that the updated version of the manuscript is approximately 90% revised compared to the original version.

1.      Calculate the turbulent statistics form the sonic (or even cups) as well and use it as reference

The updated version of the manuscript now focuses exclusively on the increased sampling rate, for which a 47-day dataset was collected. This dataset is accompanied by reference turbulence measurements provided by a sonic anemometer installed on a nearby met mast. In this revised version, we incorporate the sonic anemometer dataset for more robust comparisons and compute error metrics such as MAE, RMSE and relative to compare, along-wind variance and standard deviation derived the commercial and prototype configuration in comparison to the reference sonic anemometer measurement.

Moreover, the paper now addresses the impact of the increased sampling rate on data availability and key performance indicators (KPIs), such as the slope of the scatter plot between the mean wind speed measured by the lidar and that measured by the reference, the correlation coefficient ($R^2$) of the linear regression of the scatter plot, and the mean absolute difference in mean wind speed.

2.      Do not provide overall biases only, but also RMS error on a 10-minute basis or, even better, scatter plot like the one in Fig. 5 for lidar vs sonic

We have included the RMS error along with additional metrics such as MAE, relative error, bias, and $R^2$. Additionally, we have added a scatter plot comparing the standard deviation obtained from both the commercial and prototype lidars against sonic anemometer measurements (see Fig. 9 in the updated version).

3.      For the lidar with reduced probe volume where there is no met mast and very few data points, consider a smaller section with a lot of caution advised in the interpretation of the results

You and the first reviewer strongly recommended exercising caution when drawing conclusions related to the reduced probe length, given the very limited dataset (only 4 days) and the lack of reference measurements (e.g., from a sonic anemometer). In response to this feedback, we decided to remove the analysis on the reduced probe length. This is an ongoing work that is not ready to be published.

4.      Evaluate lidar noise also using a non-spectral approach, like the autocorrelation method by Lenschow et al., 2000 (https://doi.org/10.1175/1520-0426(2000)017<1330:MSTFOM>2.0.CO;2)

Thank you for the reference. We have implemented the ACF method and included the results in the revised manuscript. Additionally, we have compared the spectral and ACF methods for estimating noise variance (Section 3.4.1, page 15). Our analysis shows that the spectral method yields a median variance 1.5 times higher than the ACF method for the commercial lidar and twice

as high for the prototype lidar, highlighting differences in how each method characterizes noise. However, the spectral method also estimates a mean instrumental noise 30–40% lower than the ACF method, indicating variations in noise quantification. Moreover, the spectral method results in a significantly narrower spread of mean values, particularly for the commercial lidar, where the spread is reduced by half compared to the ACF method. This suggests a potential advantage in terms of consistency and stability. Based on these findings, we used the spectral method to correct the measured variance, as it provided more stable estimates of instrumental noise.

5.      The introduction could mention the effect of pulses accumulation, which is different from the sampling rate. The accumulation acts as a low-pass filter in the time domain in an analogous way as the probe average does in the spatial one. The sampling rate refers more to how quickly the lidar moves through the scan cycle, regardless of how long it takes to measure a single LOS.

We have added this text to the introduction to address your recommendation:

"The intra-beam effect generates underestimation of turbulence metrics. It arises from two anisotropic filtering processes: (1) spatial filtering due to averaging over the probe volume and (2) temporal filtering caused by averaging over the beam's pulse accumulation time, $\Delta t$, at a given measurement position. These two effects give rise to a transfer function, H, applied by the instrument on the signal measured within the probe. The transfer function includes a part due to time-averaging (the sinc term) and a part due to space-averaging (the Gaussian term), such that (e.g., Kristensen et al., 2011):" Lines 43-47, page3.

"Pulsed lidar profilers require several seconds to complete a full scanning cycle resulting in a low sampling rate that causes discrepancies between turbulence measurements taken by anemometers and those by lidar profilers (Pena et al., 2009). While the sampling rate governs how quickly the lidar progresses through a scan cycle, it is directly influenced by pulse accumulation time". Lines 58-61, page 3.

These are some modifications that would bring the paper to the standards of the other publications in the topic.

Thank you for your feedback. We have implemented all the suggested modifications to ensure the paper meets the standards of other publications in the field. The revised manuscript includes the recommended analyses, additional metrics, and methodological comparisons to enhance its rigor and clarity.

**Specific comments**

L71: "mea" instead of "mean"

Corrected.

L77: is the increased sampling rate achieved through a faster accumulation or a higher pulse repetition frequency? In the second case, the maximum range may be reduced, and it should be explained.

The increased sampling rate is reached through a reduction of pulses sent into the atmosphere. Lines 101-104, page 4.

Please add Fig. 1 angles and axis clearly indicated for readers that are unfamiliar with this technique.

We have added the positions of beams and the axis, x, y and z.

L 94: please explain what the test requirements were to consider it as "passed".

We removed this part and presented some of the test requirements. The paper now addresses the impact of the increased sampling rate on data availability and key performance indicators (KPIs), such as the slope of the scatter plot between the mean wind speed measured by the lidar and that measured by the reference, the correlation coefficient ($R^2$) of the linear regression of the scatter plot, and the mean absolute difference in mean wind speed. (Section 2.6, 2.7 and 3.1).

Equations 1 and 2: $b$ terms that should be the LOS velocities are not defined.

We defined the terms (Eq. 5-6).

Fig. 5: please add the colorbar of data density.

Done.

**Not addressed anymore in the updated version of the manuscript.**

L81: sampling rates of 0.25 Hz for wind speed may be misleading. The lidar uses a moving averaging window of 5 beams, so it does deliver a new wind speed estimate every second, but these estimates are not independent. This time overlapping effect should be made clear.

L130: the explanation of the rotation of velocity is unclear. In general, $Vx$ and $Vy$ are not 0, but after rotation $v$=0 (not $Vy$ as indicated). Aligning the x axis to North is also not the common practice in atmospheric science, where x is W-E and y is S-N, and it may be worth mentioning this as well. Please add Fig. 1 angles and axis clearly indicated for readers that are unfamiliar with this technique.

L201: it is true that the inertial subrange is limited to the right by the viscous regime where dissipation reduces TKE, but it is also limited to the left by the integral scales that supply TKE, please add this detail.

Eq 9: the | symbol to indicate the range of frequencies may be mistaken for an integration. If a fit is instead performed in this region, it would be better to remove it and explain that it is a fitting operation in the inertial subrange.

L245: is the specification of 1% relative to the error over 10 minutes or the whole dataset? Please specify.

Fig. 6: please make the box and whisker format consistent between the two subplots.

L255: have you considered that the increased difference close to the ground may be due to the lidar with reduced probe length being able to resolve better nonlinear mean wind shear?

L272: the increase in interquartile range cannot be automatically ascribed to a better sensitivity since it could very much be noise (instrumental or statistical). The fact that larger increases in standard deviation are seen at high altitude is also suspect in this sense, since one could expect the reduced probe length to lead to more recovery of turbulence variance close to the ground where length scales are smaller. If it happens at larger range, it could be noise not sensitivity.

L283: "iterative" may not be the right word, "trial and error" maybe?

L330: it is confusing saying that $\beta$=5/3 was imposed for the dissipation energy, but then $\beta$<1 were excluded. Is this a two-step process where first we fit $\beta$ to the whole spectrum, then if it passes

the check it is used for the dissipation energy with a new fit in the inertial subrange and $\beta$=5/3? Please clarify.

L359: the integral length scale is not associated with a peak in the spectrum (not premultiplied), but it is by definition its value at 0 frequency, as shown in Pope 2020, Eq. 3.114. Please remove or rephrase.

---

## Referee Report (RR1)

The current version of the manuscript "Enhancing turbulent fluctuation measurement with tailored wind lidar profilers" has been fully revised and is substantially different from the version initially submitted. The most significant change is that now only one prototype lidar with faster sampling rate is being tested against the commercial version of the WindCube 2.1 and that a great effort has been put into quantifying the effect of noise on the measurements. The narrowed scope appears reasonable and has the potential to result in a better paper.

A strength of the manuscript is the work put into a quantification of noise in the LOS data from the WindCube. Though, it is unclear why the authors did not write about the CNR value that is provided by the instrument as a standard parameter and could have been used to compare the noise levels of the prototype lidar and the commercial version.

The main weakness of the manuscript lies in the assumption that a reduction in sampling frequency leads to a reduction in variance of the measurement data, which is not true. This false assumption that increasing the sampling rate could capture additional energy associated with smaller eddies leads the interpretation of the experimental data into a wrong direction. Instead, more focus should be put on the relationship of intra beam and temporal averaging and how it is influenced by the prevailing mean wind speeds.

In its current form, the manuscript is not ready for being accepted by WES and it should be reconsidered after major revisions. Please note that the following comments are not capturing all aspects that should be improved and that a revision should be done with care before submission.

Specific comments:

Response to the reviewer:

It is good practice to acknowledge the referee's effort put into reviewing the manuscript. The authors missed this opportunity which is discouraging. Further, the response to the reviewer is suffering from mistakes, e.g., "We are right." instead of "You are right." and statements that are not covered in the updated manuscript, e.g., "We have implemented logarithmically spaced [sic] frequency bins [...] See Fig. 7b.".

1.:
The introduction gives some valuable insights into the history behind the topic, but the state of the art is insufficiently covered. Please add the most relevant and significant findings from the cited literature instead of just listing it in groups. The section should end with a guidance through the structure of the paper.

2.1:
Include a table with a comparison of the two lidar configurations showing parameters like sampling rate, accumulation time per LOS, number of samples per 30 min, range gate...

2.1:
The authors should reflect on the relationship between the industry demand for TI data (10 min.) and the variance of the u-component of the wind (30 min.) provided by the methods described in the paper.

2.2.1:
This subsubsection in the only content of subsection 2.2. This does not make sense.

2.2.1:
There are wind turbines only 210m away from the lidars, so there is no "undisturbed winds from almost all sectors". Please explain if only wind from the wind turbine's upstream direction was used in the study.

2.2.1:
Please describe the purpose of creating smaller subsets of data sampled at 0.25Hz and 1 Hz respectively. If the sonic was configured with higher sampling rate, the entire dataset could be used with 0.25Hz and 1Hz. This is unclear.

2.3:
It is wrong that Kelberlau and Mann (2020) recommend to not fit lidar-derived reconstructed velocity component data to turbulence models. They do it in their study, are satisfied with the approach and think it clarifies lidar-specific effects of turbulence sampling.

l. 245:
Provide information about the "alignment condition". What range in degrees is accepted to end with 17.1 % of the data? Is this including wind from beam 3 to beam 1, downstream of the wind turbine?

2.6:
The authors should not just claim "DNV-GL has defined acceptance criteria" but refer to the source explicitly.

2.7:
The verbal description of the quality parameters (RMSE, MAE, R2, rel. error) should be accompanied by equations that define them unambiguously.

3.2:
The description of the amount of variance included in different frequency ranges might be correct. But the conclusion that by a higher sampling rate could capture an additional percentage of the energy associated with smaller eddies is wrong. Sampling with too low frequency leads to aliasing and in a spectral display the energy from higher frequencies is folded into the lower frequency range. Instead, more focus should be put onto the relative influence of the temporal averaging caused by lower the accumulation time of the prototype lidar. Averaging does decrease the LOS variance.

3.4.1:
It is unclear why the CNR value as determined by the WindCube is not used as an indicator for the instrument noise. The median variance from spectral method for the prototype (0.0129) is also approx. 1.5 times higher than the corresponding value from the ACF method (0.0081). It is not twice as high as written in the manuscript.

Fig. 7:
The caption should be revised to explain the different purpose of subfigures (a) and (b). Also, describe which LOS direction has been used (5, vertical?)

3.5:
If the mean standard deviation is 2.9% higher, the corresponding variance must be 5.9% higher. It is unclear why the authors report 7.2%?

4:
The discussion refers to the impact of the prototype configuration on TI but it does not critically reflect on it. What happens to TI estimates if for example the v component of the turbulence wind field becomes significant, when the inflow is not aligned with one of the beams?

I suggest reducing the discussion of the potential of the prototype lidar for floating lidar systems to one sentence since floating lidar systems are not within the scope of this study.

Technical corrections:

l. 92: "True North" is wrong here because the lidar is rotated.

2.2.1: 450+1800=2256? What happened to the remaining 6 intervals?

2.6 and other occurences: DNV-GL does not exist anymore. They are named DNV now.

2.6: Refer to Table 1 and include availability thresholds.

l. 298: Replace "almost similar" by "similar"

l. 358: Replace "bin-averaged" by "wind speed-binned"

l. 408: Replace "relatively slight" by "slight"

---

## Referee Report (RR2)

**Referee's comments to wes-2024-93 – Version 2**

**General comments**

Thanks to the authors for taking the time to profoundly revise the manuscript. Narrowing down the scope and including sonic data really improved the quality of the discussion. There are minor revisions that are advised before publication.

**Specific comments**

- L37: is there a way to better define the intra-beam effect to include also the time-average correctly described next? Something like "probe-time averaging".
- Eq. 1 seems different from Eq. 19 in Kristensen et al. 2011. Please add additional references or a brief derivation.
- L214: "However, this method performs correctly only if the range in which the turbulent cascade occurs is fully captured. "Is this because of the 2/3 power law extrapolation? Lenschow shows also simpler extrapolation methods that do not require any assumption on the shape of the AFC. Please clarify.
- L221: this is the first mention of the assumption of instantaneous homogeneity. It could be better to introduce this concept earlier, possibly in the introduction, because it is fundamental to understand inter-beam contamination.
- L245: what is the tolerance around the nominal wind direction to consider it "aligned"?
- Fig. 4a: was there any consideration on the statistical or sampling error when evaluating mean wind speed profile? If statistical error bars were added to the mean profile (e.g., through bootstrapping, possibly circular) we may find out that the profiles are statistically indistinguishable. I doubt DNV does not require any statistical significance test.
- Section 3.4.1: The application of AFC requires stationary data. If this requirement was enforced, please explain how. Otherwise, clarify that the larger scattering in the AFC method could be due to the presence of non-stationarity in the data.

---

## Referee Report (RR3)

**Enhancing turbulent fluctuation measurement with tailored wind lidar profilers**

*Maxime Thiébaut, Louis Marié, Frédéric Delbos, and Florent Guinot*

**REVIEW**

GENERAL COMMENTS
1. The title seems to suggest that a new lidar can achieve substantially better turbulence measurements, however, the abstract seems to suggest that the prototype tested fell short of expectations in many ways. I suggest rephrasing the title to better match the actual outcome of the analysis.
2. In the analysis of the sonic anemometer data, have you considered (and checked for) also wake effects from the met tower structure itself? Please mention it in Section 2.2.
3. Similar to the comment above: do you expect that the structure of the tower will have an impact on the comparison between the flow measured by the sonic anemometer and each given beam from each lidar? It's a bit hard to tell from the maps, but is there a case where a beam measures the flow upwind/downwind of the met tower, while the sonic measures the opposite?
4. I am a bit confused by the practical utility of the results. The along-wind variance, on which the analysis focuses, is only one of the quantities that are used by industry to calculate TI and/or academia to calculate TKE. How do the (limited) improvements you are finding can translate to practical advancements for the calculation of TI and/or TKE? And if I am missing something and TI and TKE are not meant to be the practical utility here, what is instead?

MINOR COMMENTS
1. All statements in the first and second paragraph of the introduction (while reasonable) are missing references to substantiate the claims being made.
2. Figure 1: what do all the black dots represent in the figure? They are not explained in the caption. Also, you use both the capitalized and not-capitalized symbol for the 28-deg angle – please pick one and be consistent.

3. L. 95-105: please specify which section talks about each of the things you are listing.
4. Fig. 3: "black lines" in the caption can also represent the contour lines. Either change the color of the contours or rephrase in the caption.
5. Fig. 5: in the axis label "db" should be "dB".
6. Fig. 6: did you set the lidar such that one of its measurement heights is 97 m a.g.l.? Please specify in the paper.
7. L. 410: why do you think the % data availability may change over longer campaigns?
8. DOIs should be added to all references (whenever available) per WES standard.

---

## Author Response (AR2)

**Reviewer 1**

The current version of the manuscript "Enhancing turbulent fluctuation measurement with tailored wind lidar profilers" has been fully revised and is substantially different from the version initially submitted. The most significant change is that now only one prototype lidar with faster sampling rate is being tested against the commercial version of the WindCube 2.1 and that a great effort has been put into quantifying the effect of noise on the measurements. The narrowed scope appears reasonable and has the potential to result in a better paper.

A strength of the manuscript is the work put into a quantification of noise in the LOS data from the WindCube. Though, it is unclear why the authors did not write about the CNR value that is provided by the instrument as a standard parameter and could have been used to compare the noise levels of the prototype lidar and the commercial version.

The main weakness of the manuscript lies in the assumption that a reduction in sampling frequency leads to a reduction in variance of the measurement data, which is not true. This false assumption that increasing the sampling rate could capture additional energy associated with smaller eddies leads the interpretation of the experimental data into a wrong direction. Instead, more focus should be put on the relationship of intra beam and temporal averaging and how it is influenced by the prevailing mean wind speeds.

In its current form, the manuscript is not ready for being accepted by WES and it should be reconsidered after major revisions. Please note that the following comments are not capturing all aspects that should be improved and that a revision should be done with care before submission.

We sincerely thank the reviewer for its detailed and constructive feedback. We appreciate your recognition of the improvements made in narrowing the scope and quantifying the noise in the LOS data. Your comments have been very helpful in identifying areas that required clarification, particularly regarding the use of CNR, the assumptions around sampling frequency and variance, and the role of intra-beam and temporal averaging. We have addressed these points carefully in the revised manuscript and made substantial changes to improve both clarity and scientific accuracy.

Specific comments:

Response to the reviewer:

It is good practice to acknowledge the referee's effort put into reviewing the manuscript. The authors missed this opportunity which is discouraging. Further, the response to the reviewer is suffering from mistakes, e.g., "We are right." instead of "You are right." and statements that are not covered in the updated manuscript, e.g., "We have implemented logarithmically spaced [sic] frequency bins [...] See Fig. 7b.".

We sincerely appreciate the time and effort you dedicated to reviewing our manuscript. We deeply regret that we did not explicitly acknowledge your contribution in our initial response, and we apologize for this oversight. Your feedback is very valuable to us, and we are grateful for the constructive comments you provided.

We also apologize for the errors in our response, particularly the phrasing of "We are right" instead of "You are right." This was a mistake on our part, and we will ensure that our communication is more respectful and accurate in the future. Furthermore, we recognize that we incorrectly referenced changes that were not fully reflected in the manuscript, such as the claim about the logarithmically spaced frequency bins in Figure 7b. We have corrected these inconsistencies and updated the manuscript accordingly.

Thank you again for your thorough review and for pointing out these issues. We hope that the revised manuscript meets your expectations.

1.: The introduction gives some valuable insights into the history behind the topic, but the state of the art is insufficiently covered. Please add the most relevant and significant findings from the cited literature instead of just listing it in groups. The section should end with a guidance through the structure of the paper.

We have substantially revised the Introduction to better reflect the state of the art. Specifically, we have now included a concise summary of the most relevant and significant findings from the cited literature, rather than listing them in groups (Lines 79-89, page 4). Additionally, as recommended, we have added a brief paragraph at the end of the Introduction to guide the reader through the structure of the paper (Lines 96-104, page 4).

2.1: Include a table with a comparison of the two lidar configurations showing parameters like sampling rate, accumulation time per LOS, number of samples per 30 min, range gate…

We have added Table 1 (page 6) to the manuscript, which provides a side-by-side comparison of the two lidar configurations. The table includes key parameters such as the sampling rate, accumulation time per LOS, number of samples per 30 minutes, and probe length.

2.1: The authors should reflect on the relationship between the industry demand for TI data (10 min.) and the variance of the u-component of the wind (30 min.) provided by the methods described in the paper.

We have added this in the Introduction: "This enhancement is assessed for its impact on measuring mean wind speed, data availability, and along-wind variance and its square root, i.e., the standard deviation. The latter is particularly important, as it is used in the wind power industry to compute turbulence intensity (TI), a critical metric for turbine load assessment, site suitability, and energy yield predictions." (Lines 91-94, page 4), and this:

"The selection of a 30-min window, rather than the standard 10-minute interval commonly used in the wind energy industry, was guided by the aim of reducing random errors in turbulence measurements, following the recommendations of Lenschow et al. (1994)." (Lines 165-167, page 8).

2.2.1: This subsubsection is the only content of subsection 2.2. This does not make sense.

As suggested, we have removed subsubsection 2.2.1, since it was the only content within subsection 2.2, making the subdivision unnecessary. The content now appears directly under subsection 2.2 for improved clarity and structure.

2.2.1: There are wind turbines only 210 m away from the lidars, so there is no "undisturbed winds from almost all sectors". Please explain if only wind from the wind turbine's upstream direction was used in the study.

We have added a wind rose to illustrate the sector contaminated by the turbine wake, which is highlighted by the blue shaded areas in Fig. 4 (page 8). The wind sectors selected for the present analysis, indicated by the gray shaded areas in the same figure, were carefully chosen to lie outside the contaminated region.

2.2.1: Please describe the purpose of creating smaller subsets of data sampled at 0.25Hz and 1 Hz respectively. If the sonic was configured with higher sampling rate, the entire dataset could be used with 0.25Hz and 1Hz. This is unclear.

To clarify, only the sonic anemometer data were resampled to 1 Hz in order to match the sampling rate of the prototype lidar. The datasets from the commercial and prototype lidars themselves were not resampled; their native sampling rates of 0.25 Hz and 1 Hz, respectively, were retained. Consequently, each 30-minute subset contained 450 data points for the commercial lidar and 1,800 data points for the prototype lidar. This clarification has been added to the revised manuscript (Lines 163–165, page 8).

2.3: It is wrong that Kelberlau and Mann (2020) recommend to not fit lidar-derived reconstructed velocity component data to turbulence models. They do it in their study, are satisfied with the approach and think it clarifies lidar-specific effects of turbulence sampling.

We removed this part.

l. 245: Provide information about the "alignment condition". What range in degrees is accepted to end with 17.1 % of the data? Is this including wind from beam 3 to beam 1, downstream of the wind turbine?

It was ± 5°. We have added this sentence in the revised version: "In this paper, we restrict the application of the variance method to situations where the wind aligns (± 5°) with a single pair of opposite beams (either pair 1-3 or pair 2-4) of the lidar profilers. (Lines 265-267, page 11).

2.6: The authors should not just claim "DNV-GL has defined acceptance criteria" but refer to the source explicitly.

We have updated the manuscript to explicitly reference the source of the acceptance criteria defined by DNV GL, rather than simply stating that "DNV-GL has defined acceptance criteria." We have made this change throughout the manuscript wherever DNV GL is mentioned.

2.7: The verbal description of the quality parameters (RMSE, MAE, R2, rel. error) should be accompanied by equations that define them unambiguously.

These equations can now be found as Equations 7–10 on pages 12–13.

3.2: The description of the amount of variance included in different frequency ranges might be correct. But the conclusion that by a higher sampling rate could capture an additional percentage of the energy associated with smaller eddies is wrong. Sampling with too low frequency leads to aliasing and in a spectral display the energy from higher frequencies is folded into the lower frequency range. Instead, more focus should be put onto the relative influence of the temporal averaging caused by lower the accumulation time of the prototype lidar. Averaging does decrease the LOS variance.

As recommended, we have revised the manuscript to place greater emphasis on the influence of temporal averaging resulting from pulse accumulation time. Specifically, we now explain that the higher variance observed with the prototype lidar is primarily attributed to its shorter accumulation time, which reduces temporal averaging and better preserves along-wind variance. Additionally, we clarify that the shorter accumulation time also limits the advection-driven increase in effective probe length, thereby reducing the impact of spatial averaging, particularly at higher wind speeds. These revisions have been consistently implemented throughout the manuscript, including in the abstract, data and methods, results, discussion, and conclusion sections.

3.4.1: It is unclear why the CNR value as determined by the WindCube is not used as an indicator for the instrument noise. The median variance from spectral method for the prototype (0.0129) is also approx. 1.5 times higher than the corresponding value from the ACF method (0.0081). It is not twice as high as written in the manuscript.

Mean CNR profiles have been added to Fig. 5b (page 14) and discussed in Section 3.4.1 (lines 359–364, page 17). We also thank the reviewer for identifying an error in the median variance reported for the spectral method. As correctly noted, it is approximately 1.5 times higher than the corresponding value from the ACF method. This correction has been made in the revised manuscript (line 366, page 17).

Fig. 7: The caption should be revised to explain the different purpose of subfigures (a) and (b). Also, describe which LOS direction has been used (5, vertical?)

Fig. 7 is now Fig. 8. Here is the new title: "(a) LOS velocity spectrum measured by beam 5 of the prototype lidar (solid black), fitted using Eq. 2 with three different weighting schemes: unweighted (dashed green), low-frequency weighted (dashed red), and high-frequency weighted (dashed blue). This panel corresponds to the study focused on selecting the optimal weighting scheme. (b) The optimal scheme (high-frequency weighted) is applied to LOS velocity spectrum measured by beam 5 of the commercial lidar (blue) and the prototype lidar (orange).", page 16.

3.5: If the mean standard deviation is 2.9% higher, the corresponding variance must be 5.9% higher. It is unclear why the authors report 7.2%?

Sorry, we have made a mistake. The mean variance of 7.2% is correct but the mean standard deviation is wrong. The true mean standard deviation is 3.5% (Line 407, page 20).

4: The discussion refers to the impact of the prototype configuration on TI but it does not critically reflect on it. What happens to TI estimates if for example the v component of the turbulence wind field becomes significant, when the inflow is not aligned with one of the beams?

We have decided to remove the discussion regarding the impact of the prototype configuration on TI, as the original computation focused on specific wind directions. This approach is too restrictive and does not adequately represent the variability encountered in practical wind power applications. As such, it does not provide relevant or generalizable insights for the broader wind energy industry. We have therefore omitted this paragraph to maintain the focus on more representative and applicable results.

I suggest reducing the discussion of the potential of the prototype lidar for floating lidar systems to one sentence since floating lidar systems are not within the scope of this study.

We agree with the reviewer that floating lidar systems are outside the scope of this study. Since it proved difficult to condense the discussion of the prototype's potential for floating lidar systems into a single sentence that integrates well with the surrounding paragraphs, we have decided to remove the paragraph entirely to maintain clarity and focus.

Technical corrections:

l. 92: "True North" is wrong here because the lidar is rotated.

You are right. We have modified this.

2.2.1: 450+1800=2256? What happened to the remaining 6 intervals?

This is 2256 30-min subsets. For the commercial and lidar each subset contains 450 measurement points. For the prototype lidar, each subset contains 1,800 measurements. We have modified this part to make it clearer for the reader. (Lines 163–165, page 8).

2.6 and other occurrences: DNV-GL does not exist anymore. They are named DNV now.

Each occurrence has been corrected.

2.6: Refer to Table 1 and include availability thresholds.

This table is now Table 2 which includes information on availability threshold. We now refer to this Table in the text (line 278 and line 281, page 12).

l. 298: Replace "almost similar" by "similar"

Done

l. 358: Replace "bin-averaged" by "wind speed-binned"

Done

l. 408: Replace "relatively slight" by "slight"

Done

**Reviewer 2**

**General comments**

Thanks to the authors for taking the time to profoundly revise the manuscript. Narrowing down the scope and including sonic data really improved the quality of the discussion. There are minor revisions that are advised before publication.

We sincerely thank the reviewer for their thoughtful and constructive feedback throughout the review process. We appreciate your recognition of the improvements made, particularly regarding the inclusion of sonic data and the refined scope. Your comments have been very valuable in enhancing the clarity and quality of the manuscript.

**Specific comments**

• L37: is there a way to better define the intra-beam effect to include also the time-average correctly described next? Something like "probe-time averaging".

Thank you for your suggestion, "probe-time averaging" is indeed appropriate. We have incorporated this term into the revised version: "The intra-beam effect refers to a probe-time averaging phenomenon occurring within the lidar probe, leading to an underestimation of turbulence metrics." (Line 45, page 3).

• Eq. 1 seems different from Eq. 19 in Kristensen et al. 2011. Please add additional references or a brief derivation.

You are correct, Eq. 1 is indeed different from Eq. 19 in Kristensen et al. (2011). We now include the full mathematical derivation of Eq. 1 in the supplementary material to clarify this distinction.

• L214: "However, this method performs correctly only if the range in which the turbulent cascade occurs is fully captured. "Is this because of the 2/3 power law extrapolation? Lenschow shows also simpler extrapolation methods that do not require any assumption on the shape of the AFC. Please clarify.

Thank you for the comment. We chose to focus on the power-law fit approach, as it aligns with our analysis framework. However, to address concerns about the validity of this method, we now include a stationarity test in the revised manuscript to support the applicability of the ACF-based approach.

• L221: this is the first mention of the assumption of instantaneous homogeneity. It could be better to introduce this concept earlier, possibly in the introduction, because it is fundamental to understand inter-beam contamination.

Thank you. In the revised the version, we now mention the instantaneous homogeneity in the introduction: "This effect is particularly relevant in the context of the

assumption of instantaneous homogeneity, which underlies multi-beam lidar measurement techniques" (Lines 39-40, page 2).

• L245: what is the tolerance around the nominal wind direction to consider it "aligned"?

The tolerance was ± 5°. We have added this sentence in the revised version: "In this paper, we restrict the application of the variance method to situations where the wind aligns (± 5°) with a single pair of opposite beams (either pair 1-3 or pair 2-4) of the lidar profilers (Lines 265-267, page 11). Also, we have added a wind rose to illustrate the wind sectors selected for the present analysis, indicated by the gray shaded areas in Fig. 4 of the revised version.

• Fig. 4a: was there any consideration on the statistical or sampling error when evaluating mean wind speed profile? If statistical error bars were added to the mean profile (e.g., through bootstrapping, possibly circular) we may find out that the profiles are statistically indistinguishable. I doubt DNV does not require any statistical significance test.

Thank you for your comment. We have updated the figure (now Fig. 5a) to include error bars representing the 95% confidence interval, calculated using bootstrapping. This addition illustrates the statistical uncertainty of the mean wind speed profile and allows for a more robust interpretation of potential differences between profiles.

• Section 3.4.1: The application of AFC requires stationary data. If this requirement was enforced, please explain how. Otherwise, clarify that the larger scattering in the AFC method could be due to the presence of non-stationarity in the data.

Thank you for your comment. We have added a stationarity test using the Augmented Dickey-Fuller (ADF) method to assess the stationarity of each 30-minute subset and, consequently, the validity of the ACF-based approach. This test is now described in Section 2.4.2 (Lines 234–237, Page 10), and the results are presented in Section 3.4.2 (Lines 373–379, Pages 17–18).

---

## Author Response (AR3)

Dear Editor,

We have revised the manuscript in accordance with the reviewers' comments. We have carefully addressed all of Reviewer 3's suggestions, which we found constructive and valuable for improving the quality of our work. Additionally, we have corrected the technical issue raised by Reviewer 2 regarding the missing reference to Frehlich (1994)a, which is now properly included in the bibliography.

In the present document, we respond point by point to the comments from Reviewer 3 (in black). Our responses are provided in blue.

GENERAL COMMENTS

1. The title seems to suggest that a new lidar can achieve substantially better turbulence measurements, however, the abstract seems to suggest that the prototype tested fell short of expectations in many ways. I suggest rephrasing the title to better match the actual outcome of the analysis.

Thank you for your comment. We agree that the original title may have implied a level of performance improvement not fully supported by the results presented. To better reflect the scope and findings of our study, we have revised the title to: **"Evaluating enhanced sampling rate for turbulence measurement with wind lidar profiler."** This revised title emphasizes the focus on evaluating the impact of increased sampling rate rather than suggesting definitive performance gains, aligning more closely with the content and conclusions of the manuscript.

2. In the analysis of the sonic anemometer data, have you considered (and checked for) also wake effects from the met tower structure itself? Please mention it in Section 2.2.

Yes, we considered potential wake effects from the meteorological mast structure in our analysis of the sonic anemometer data. The wind direction sectors affected by mast-induced flow disturbances were found to overlap with those influenced by wind turbine WT N117. These sectors were excluded from the turbulence analysis, as shown in the blue regions of Fig. 4. We have added these sentences to the text: "Potential wake effects from the meteorological mast structure were considered in the analysis of the sonic anemometer data. The wind directions associated with flow disturbances caused by the mast itself overlap with the wake sector of wind turbine WT N117 which was excluded from the analysis." (Lines 155-157, page 7).

3. Similar to the comment above: do you expect that the structure of the tower will have an impact on the comparison between the flow measured by the sonic anemometer and each given beam from each lidar? It's a bit hard to tell from the maps, but is there a case where a beam measures the flow upwind/downwind of the met tower, while the sonic measures the opposite?

We also took care to avoid mismatches between lidar beam positions and the sonic anemometer due to potential obstruction by the mast. The orientation of the lidar beams was selected to avoid cases where the lidar measured flow either directly upwind or downwind of the mast relative to the sonic.

4. I am a bit confused by the practical utility of the results. The along-wind variance, on which the analysis focuses, is only one of the quantities that are used by industry to calculate TI and/or academia to calculate TKE. How do the (limited) improvements you are finding can translate to practical advancements for the calculation of TI and/or TKE? And if I am missing something and TI and TKE are not meant to be the practical utility here, what is instead?

You are correct that along-wind variance is only one component in the calculation of TI and TKE. In this study, we focused on evaluating the performance of an enhanced-sampling-rate lidar specifically in capturing the along-wind variance because it is typically the dominant component of TKE in atmospheric surface-layer flows and is often the most reliably measured by profiling lidars, given their beam configuration and scanning limitations.

While we do not claim that the observed improvements fully resolve the limitations of current lidars in estimating TI or TKE, our results provide a targeted assessment of how increasing the sampling rate affects the accuracy of a critical turbulence parameter. This is a necessary step toward better characterizing the capabilities of lidar systems for advanced turbulence measurements.

The practical utility of our work lies in identifying the potential and the limitations of using enhanced-sampling-rate lidar systems for future applications that require finer temporal resolution, such as site assessments for wind energy projects, model validation, and inflow condition characterization. Ultimately, the insights gained here can inform both lidar system design and the development of correction or filtering techniques for TI and TKE estimation.

We mention TI in the introduction: "This enhancement is assessed for its impact on measuring mean wind speed, data availability, and along-wind variance and its square root, i.e., the standard deviation. The latter is particularly important, as it is used in the wind power industry to compute turbulence intensity (TI), a critical metric for turbine load assessment, site suitability, and energy yield predictions." (Lines 95-97, page 4).

MINOR COMMENTS

1.      All statements in the first and second paragraph of the introduction (while reasonable) are missing references to substantiate the claims being made.

We have added five references in the first two paragraphs.

2. Figure 1: what do all the black dots represent in the figure? They are not explained in the caption. Also, you use both the capitalized and not-capitalized symbol for the 28-deg angle – please pick one and be consistent.

We have added the meaning of the black dots in the title of Fig. 1: "The black dots indicate the centers of the probe measurement volumes.". Moreover, we choose to use the not-capitalized symbol throughout the paper.

3. L. 95-105: please specify which section talks about each of the things you are listing.

We have now specified the corresponding sections.

4. Fig. 3: "black lines" in the caption can also represent the contour lines. Either change the color of the contours or rephrase in the caption.

We now used the word "arrows" instead of "lines".

5. Fig. 5: in the axis label "db" should be "dB".

The label has been changed.

6. Fig. 6: did you set the lidar such that one of its measurement heights is 97 m a.g.l.? Please specify in the paper.

Yes, we have added this information to the text: "One of the measurement heights of both lidars was set to 97 m above ground to coincide with the height of the sonic anemometer deployment on the mast." (Lines 162-163, page 7).

7. L. 410: why do you think the % data availability may change over longer campaigns?

This is a good remark. We now specified the reason: "While this difference is minimal, longer measurement campaigns, typically lasting over a year for wind site characterization, may accumulate more instances of data loss due to environmental factors, hardware limitations, or maintenance events, potentially making the impact of reduced availability more noticeable over time." (Lines 415-418, pages 20-21).

8. DOIs should be added to all references (whenever available) per WES standard

We have added the DOI for all the references when available.